

SciPost Phys. Lect. Notes 65 (2022)

# Seven *Études* on dynamical Keldysh model

**Dmitri V. Efremov[1] and Mikhail N. Kiselev[2]⋆**

**1** IFW Dresden, Helmholtzstr. 20, 01069 Dresden, Germany
**2** The Abdus Salam International Centre for Theoretical Physics,
Strada Costiera 11, I-34151, Trieste, Italy

⋆ mkiselev@ictp.it

## Abstract

We present a comprehensive pedagogical discussion of a family of models describing the propagation of a single particle in a multicomponent non-Markovian Gaussian random field. We report some exact results for single-particle Green's functions, self-energy, vertex part and T-matrix. These results are based on a closed form solution of the Dyson equation combined with the Ward identity. Analytical properties of the solution are discussed. Further we describe the combinatorics of the Feynman diagrams for the Green's function and the skeleton diagrams for the self-energy and vertex, using recurrence relations between the Taylor expansion coefficients of the self-energy. Asymptotically exact equations for the number of skeleton diagrams in the limit of large $N$ are derived. Finally, we consider possible realizations of a multicomponent Gaussian random potential in quantum transport via complex quantum dot experiments.

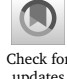

# 1 *Prelude.* Keldysh model

The original Keldysh Model (KM) was proposed by Keldysh in 1965 [1]. He showed that
a single-particle problem of electron's propagation in a single-component random Gaussian
field $V(r)$ with forward scattering which is given by the correlator

$$D(\mathbf{r}-\mathbf{r}') = \langle V(\mathbf{r})V(\mathbf{r}')\rangle = W^2 \rightarrow D(\mathbf{q}) = (2\pi)^3 W^2 \delta(\mathbf{q}), \tag{1}$$

can be solved exactly by the Feynman diagrammatic technique.

   The idea behind Keldysh's solution is as follows. According to the diagrammatics rules, the
diagram of order $N$ contains $N$ lines of the Gaussian random field (denoted by wavy lines),
$N + 2$ Green's functions (denoted dy solid lines) and $2N$ vertices [2]. The total number of
diagrams is given by the number $A_N$ corresponding to the total number of possibilities to
connect $2N$ vertices with $N$- lines:

$$A_N = (2N-1)!! = \frac{(2N-1)!}{2^{N-1}(N-1)!}. \tag{2}$$

An example of the diagrammatic expansion corresponding to the single electron Green's func-
tion $G$ is shown in Fig.1.

   For the chosen Gaussian random field Eq.(1) all Feynman graphs in the order $N$ give the
same contribution to the Green's function (GF). As a result, the GF is represented by the fol-
lowing form:

$$G(E) = G_0(E)\left\{1 + \sum_{N=1}^{\infty}(2N-1)!! G_0^{2N}(E)W^{2N}\right\}. \tag{3}$$

Here we use the shorthand notation $E = \epsilon - \mathbf{p}^2/(2m)$ and $G_0(E) = 1/E$ is the bare Green
function. Using the representation for the Euler Gamma-function:

$$(2N-1)!! = \frac{1}{\sqrt{2\pi}}\int_{-\infty}^{\infty} dt\, t^{2N-2} e^{-t^2/2}, \tag{4}$$

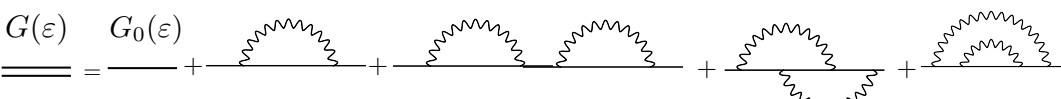

Figure 1: Diagrammatic expansion for the Green's function. The double line corre-
sponds to bold Green's function. The single lines denote bare Green's functions. The
wavy lines stand for the correlator of the Gaussian classical random field.

and summing up the geometric progression results in a very simple expression for the full GF:

$$G^R(E) = \frac{1}{\sqrt{2\pi W^2}} \int_{-\infty}^{\infty} dV \frac{e^{-V^2/2W^2}}{E - V + i\delta} \cdot \tag{5}$$

This equation has a very simple physical meaning. Propagation of an electron in a Gaussian random field corresponds to an ensemble Gaussian averaging of the single particle Green's function over Gaussian realizations of the random potential. This is so-called zero-dimensional realization of the Gaussian disorder.

The next very important contribution to the physics of the Keldysh model was made by Efros in [3]. He built a theory of semiconductors with very shallow charge impurities located at random positions. The potential created by the charge impurities is the screened Coulomb potential characterized by the screening radius $r_0$. If the electron's density $n_0$ satisfies the condition $n_0 r_0^3 \gg 1$, then the electrons experience the potential only at the point at which they are located. The correlations of the impurity potential can be represented as a superposition of the short-scale and large-range parts. The large-scale fluctuations are exactly accounted in the Gaussian field approximation. The short-range correlations are accounted for by the perturbation theory. It was shown that the single particle Green's function satisfies the Dyson equation which for this model becomes an ordinary differential equation:

$$W^2 \frac{dG(E)}{dE} + E \cdot G(E) = 1. \tag{6}$$

The Dyson equation is obtained using the Ward identity [3], which has a very simple form due to $\delta(\mathbf{q})$-shape of the Gaussian random potential correlator. One of the direct consequences of the Keldysh-Efros theory is the emergence of a tail in the single-particle density of states (DOS) at $\epsilon < 0$. The formation of such tails in DOS is one of the general properties of disordered systems. For a detailed description of the original Keldysh model and its properties we refer to a remarkable book by M.V. Sadovskii [4].

Recently, it has been suggested that the Keldysh model can be realized not only in systems with long-range *spatial* variations of disordered potentials, but also in a system characterized by slow *temporal* fluctuations of the confining potential [6]. Such behaviour can be realized, for example, in functional nano-structures [7]. Generalized dynamical Keldysh models pave the way towards understanding of systems characterized by multi-component Gaussian non-Markovian random fields. A few-component dynamical Keldysh model have been proposed in [6–8] to describe effects of a slow electric gate noise in double quantum dots. In this paper, we develop the ideas of [6] in connection with implementation of the multi-component Keldysh model in the complex quantum dots.

These Notes are organized as follows: In *Intermezzo* we discuss the realizations of dynamical (time-dependent) multi-component Keldysh models in quantum circuits containing multiple quantum dots. We demonstrate that fluctuations of two potentials: the confinement potential and the inter-nodal barrier potential, which arise due to fluctuations of the corresponding electric gates, can create all the necessary ingredients for a multicomponent Keldysh model. The first *Étude* is a very simple exercise detailing the "orthodox" single-component Keldysh model; the second *Étude* uses the same "theme" to extend the analysis to a two-component non-Markovian random Gaussian field; the third *Étude* is devoted to the three-component model and contains two variations involving two different realizations of a three component slow noise model; the fourth short *Étude* summarizes the technique studied in all previous *Études*. The fifth *Étude* begins a new melody of Feynman diagram combinatorics; the sixth and seventh *Études*, being very advanced, are devoted to enumeration of skeleton diagrams for the one-component model (*Étude* 6) including a new derivation of the Sadovskii-

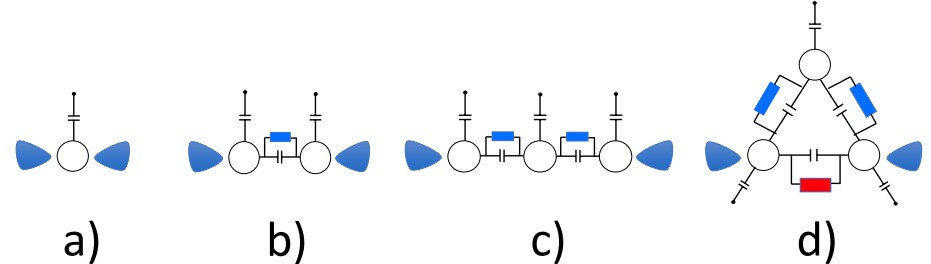

Figure 2: Experimental realization of the Keldysh model in quantum dots: a) single QD; b) serial double QD; c) linear (serial) triple QD; d) triangular triple QD.

Kuchinskii-Suslov equation and the multi-component models ($\acute{E}tude$ 7) described by the generalized recurrence equation. The necessary mathematical instrumentation are presented in the Appendices.

## 2  *Intermezzo*. Realization of dynamical multi-component Keldysh models with complex quantum dots

The goal of this Section is to discuss possible realizations of time-dependent Keldysh models in functional nano-structure devices (see Fig. 2 for an illustration). The key elements of the electric circuit consist of quantum dots (see details below), gates controlling the shape of confining potential, gates controlling the shape of the inter-dot barriers and a global back gate operating uniformly on the dot's levels. The nano-devices are connected by the leads (contacts) to the rest of the quantum circuit.

**Single QD**

We start with a single quantum dot (QD) Fig. 2a which is an island of two- or three- dimensional electron gas confined by an electrostatic potential (see Fig. 3). We assume that QD operates in the Coulomb Blockade regime [7]. There is a global noisy gate attached to QD. This gate slowly changes the confining potential.

The single-particle Hamiltonian describing the dot (we do not discuss the contacts attached to it) is given by the following expression:

$$H = [\epsilon_0 + \lambda] n. \tag{7}$$

Here $n = c^\dagger c$ with $c$ being annihilation operator of the single electron state in QD. For simplicity we consider spinless (spin polarized) electrons. We refer to [7] for the Coulomb blockade

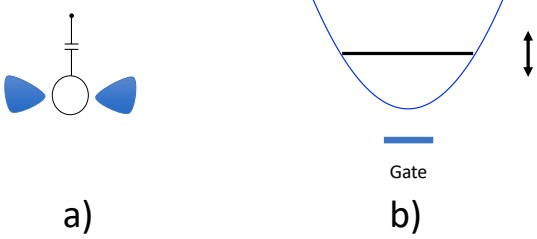

Figure 3: Left: single quantum dot with the noisy back gate. Right: quantum well with fluctuating energy level.

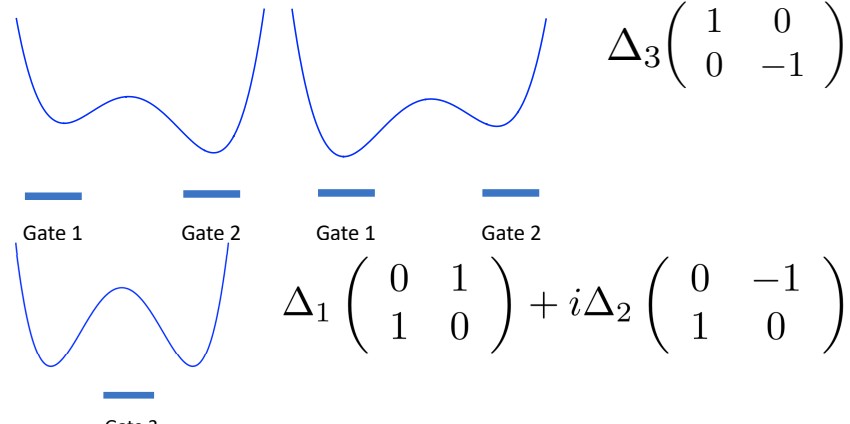

Figure 4: Slow noise produced by the back gate in double QD. Top: "diagonal" noise describing fluctuations of the level positions; Bottom: $U(1)$ fluctuations of the barrier. Three $2 \times 2$ Pauli matrices constitute the basis for the fundamental representation of the $SU(2)$ group.

effects. The classical Gaussian random potential $\lambda$ is defined by its mean value and the two-point correlation function:

$$\langle \lambda(t) \rangle = 0, \quad \langle \lambda(t)\lambda(t') \rangle = D_1(t - t'). \tag{8}$$

We use a color noise to describe the two-point correlator of the classical Gaussian noise.

$$D_1(t - t') = W^2 e^{-\gamma |t - t'|}, \tag{9}$$

where $\gamma = 1/t_{\text{noise}}$ defines the characteristic time for the noise correlation function and $W$ is the amplitude of the noise. There are two important limits:

$$\gamma \rightarrow \infty : \quad D_1(t - t') \rightarrow \delta(t - t'), \tag{10}$$

$$\gamma \rightarrow 0 : \quad D_1(\omega) \rightarrow \delta(\omega). \tag{11}$$

Here $D_1(\omega)$ is the Fourier transform of $D_1(t - t')$. The equation corresponds to a fast noise (it's "fastest" realization is given by the white noise) and describes Markovian processes. The second equation corresponds to the a slow noise (it's "slowest" realization is known as the Keldysh model) representing the Gaussian disorder with a long (infinite) memory. In $\acute{E}tudes$ we focus on a discussion of the physics described by the Keldysh model.

The single particle Green's function in given realization of the quasi-static disorder is:

$$G^R(\epsilon) = \frac{1}{\epsilon - \epsilon_0 - \lambda + i\delta}. \tag{12}$$

Averaging over the realizations of the Gaussian disorder can be done equivalently by means of the distribution function $P(\lambda) = 1/\sqrt{2\pi W^2} \exp(-\lambda^2/(2W^2))$.

## Double QD

The single-particle Hamiltonian describing a two-level system (double well potential, double quantum dot Fig. 2b) consists of two terms: the diagonal $H_Z$ and the off-diagonal (tunneling) part $H_{\text{tun}}$. The diagonal part is defined as follows:

$$H_Z = [(\epsilon_0 + \lambda)] n_\alpha + \Delta_3 S^z, \tag{13}$$

where $\alpha = l, r$ stand for the left and right quantum well respectively, $S^z = n_l - n_r$. Thus, the diagonal part consists of a uniform term acting on the levels of each dot similar to that discussed in the previous subsection and an alternating term acting similarly to the longitudinal random synthetic magnetic field (see Fig.4 upper panel)[1]. Since the effects of the uniform noise $\lambda$ have been discussed above, in this subsection we focus on the effects of the synthetic magnetic field. The correlator of the longitudinal random Gaussian potential is given by:

$$\langle \Delta_3(t) \rangle = 0, \quad \langle \Delta_3(t)\Delta_3(t') \rangle = D_1(t - t').$$
(14)

The off-diagonal (tunneling) Hamiltonian accounting for the tunneling through the barrier separating the two quantum wells (see Fig.4 lower panel) is given by:

$$H_{\text{tun}} = \Delta S^- + \Delta^* S^+,$$
(15)

where $\Delta = \Delta_1 + i\Delta_2$. We introduced the pseudo-spin operators $S^+ = c_r^\dagger c_l$, $S^- = c_l^\dagger c_r$ such that the tunneling from one quantum well to another represents the pseudo-spin flip process. The two-component Gaussian random field describing slow non-Markovian fluctuations of the barrier is defined as:

$$\langle \Delta(t) \rangle = 0, \quad \langle \Delta(t)\Delta^*(t') \rangle = D_2(t - t'),$$
(16)

with

$$D_2(t - t') = 2W^2 e^{-\gamma|t-t'|}.$$
(17)

Thus, the Hamiltonian describing the double well potential is expressed in terms of the basis of the Pauli matrices - the fundamental representation of the group $SU(2)$.[2] The single particle Green's function in a given realization of the off-diagonal disorder (only) is:

$$G_l^R(\epsilon) = G_r^R(\epsilon) = \frac{\epsilon - \epsilon_0}{(\epsilon - \epsilon_0 + i\delta)^2 - |\Delta|^2}.$$
(19)

The lines $|\Delta|^2 = \Delta_1^2 + \Delta_2^2 = const$ define circles $S_1$ in the $d = 2$ parametric space.

The single particle Green's functions in the presence of both the diagonal and the off-diagonal disorders are given by:

$$G_{r/l}^R(\epsilon) = \frac{\epsilon - \epsilon_0 \pm \Delta_3}{(\epsilon - \epsilon_0 + i\delta)^2 - \Delta_3^2 - |\Delta|^2}.$$
(20)

The surfaces $|\Delta|^2 + \Delta_3^2 = \Delta_1^2 + \Delta_2^2 + \Delta_3^2 = const$ define spheres $S_2$ in the $d = 3$ parametric space. When both uniform and staggered diagonal noises are present in addition to the off-diagonal noise, the GF in the given realization of quasi-static (infinite memory) Gaussian random potential casts the form:

$$G_{r/l}^R(\epsilon) = \frac{\epsilon - \epsilon_0 - \lambda \pm \Delta_3}{(\epsilon - \epsilon_0 - \lambda + i\delta)^2 - \Delta_3^2 - |\Delta|^2}.$$
(21)

---

[1]There is no real magnetic field in the model since all fields are associated with noisy electric gates.

[2]The sum of the Hamiltonians (13) and (15) can be written as

$$H = \left[ (\epsilon_0 + \lambda)\tau^0 + \Delta_3\tau^z + \Delta\tau^- + \Delta^*\tau^+ \right]_{\alpha\beta} c_\alpha^\dagger c_\beta,$$
(18)

where $\tau^i$ are the Pauli matrices ($i = x, y, z$) and $\tau^0$ is the unit $2 \times 2$ matrix.

## Triple QD

Triple quantum dots (TQD) consist of three electron gas islands and either two (serial TQD, Fig. 2c) or three (triangular TQD, Fig. 2d) barriers separating the QDs. For the purposes of our *Études* we consider a *serial* TQD only to avoid a discussion of magnetic flux effects. The triangular TQD will be considered elsewhere. A cartoon illustrating the triple well potentials is shown in Fig. 5.

A natural basis for representing single-electron processes in the serial triple quantum dots is the basis of eight Gell-Mann matrices - the fundamental representation of the $SU(3)$ group (see Appendix A). The classification of all possible contributions to the Hamiltonian is shown on Fig. 5. Note, that we use a "rotated" representation of the Gell-Mann matrices [9] different

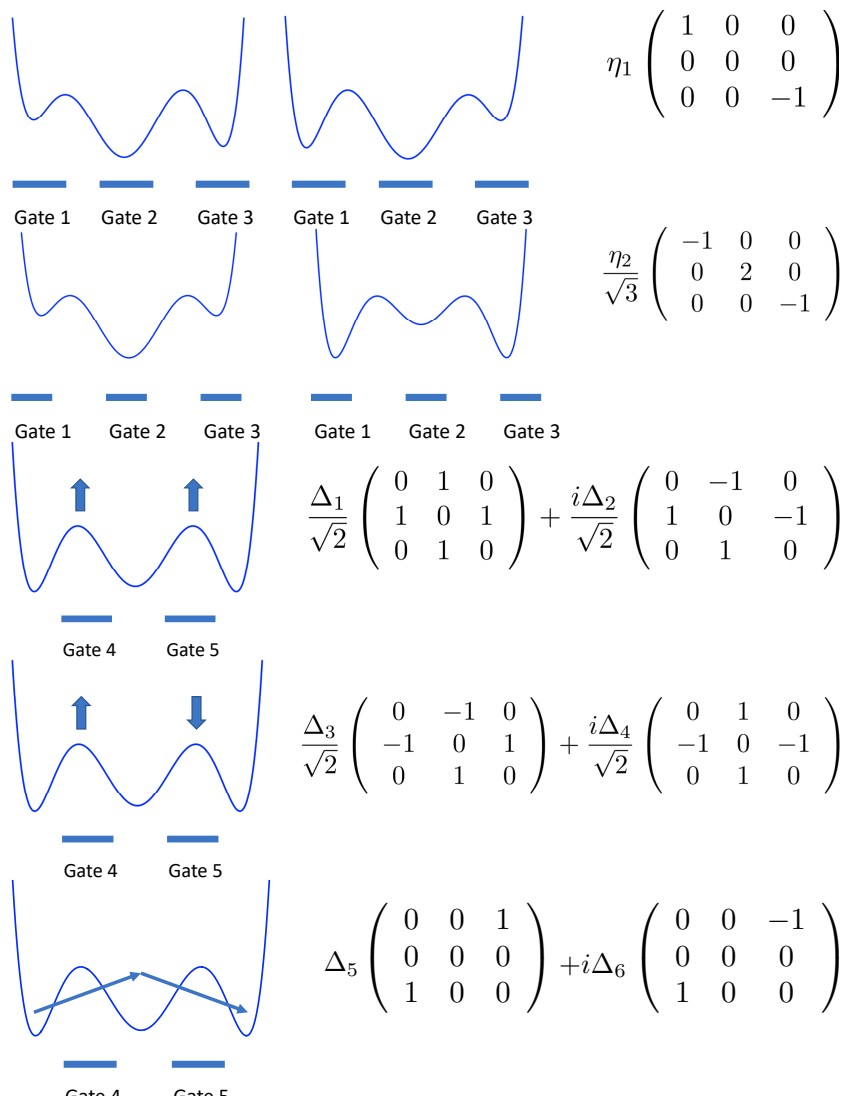

Figure 5: Slow noise produced by the back gate in the serial triple QD. First and second rows: fluctuations of the level positions in each of three QD. Third row: symmetric fluctuations of the barriers. Fourth row: anti-symmetric fluctuations of the barriers. Bottom row: fluctuations of the co-tunneling. The same processes will naturally appear in the triangular QD Fig. 2d and describe direct tunneling between the first and the third dot. Eight $3 \times 3$ Gell-Mann matrices constitute the basis for the fundamental representation of the $SU(3)$ group.

from the conventional $SU(3)$ basis. This basis embeds three $SU(2)$, $S = 1$ generators as the first three Gell-Mann matrices (see Appendix A). The last five Gell-Mann matrices represent quadrupole moments expressed as bilinear combinations of $S^x$, $S^y$ and $S^z$ [9].

First, there are two types of the non-uniform diagonal processes described by two diagonal Gell-Mann matrices $\mu_3$ and $\mu_8$ (see Appendix A)[3] forming the Cartan basis:

$$H_{\text{diag}} = \left[\epsilon_0 + \eta_1(t)\mu_3^\alpha + \eta_2(t)\mu_8^\alpha\right]n_\alpha. \tag{22}$$

We introduce two scalar random Gaussian fields $\eta_1$ and $\eta_2$ to account for the diagonal noise (see first two rows in Fig. 5).

The off-diagonal random processes can be classified as follows:

i) Processes which can be considered as independent tunneling processes in the double well potential. These terms describe slow symmetric (in-phase) fluctuations of the two barriers (see the third row in Fig. 5):

$$H_1 = \Delta_{12}S^- + \Delta_{12}^*S^+. \tag{23}$$

Here $\Delta_{12} = \Delta_1 + i\Delta_2$, $\hat{S}^\pm = (\hat{S}^x \pm i\hat{S}^y)/\sqrt{2}$ and $S^+ = c_r^\dagger c_c + c_c^\dagger c_l$, $(S^+)^\dagger = S^-$ (see Appendix A), and we label the states in the dot $c_\alpha$ with $\alpha = l, c, r$ for the left, central and right quantum well correspondingly.

ii) Processes which can be considered as anti-symmetric (out-of-phase) fluctuations of the barriers (see the fourth row in Fig. 5).

$$H_2 = \Delta_{34}T^- + \Delta_{34}^*T^+. \tag{24}$$

Here $\hat{T}^\pm = (\hat{T}^x \pm i\hat{T}^y)/\sqrt{2}$, $T^+ = c_c^\dagger c_l - c_r^\dagger c_c$, $(T^+)^\dagger = T^-$ (see Appendix A). We defined the two-component random Gaussian field as $\Delta_{34} = \Delta_3 + i\Delta_4$.

iii) Processes directly connecting the left and right quantum wells (see fifth and the last row in Fig. 5).

$$H_3 = \Delta_{56}P^- + \Delta_{56}^*P^+. \tag{25}$$

Here $\hat{P}^\pm = (\hat{P}^x \pm i\hat{P}^y)/2$, and $P^+ = c_r^\dagger c_l$, $(P^+)^\dagger = P^-$ (see Appendix A). We define the two-component random Gaussian field as $\Delta_{56} = \Delta_5 + i\Delta_6$. For a linear TQD such processes require co-tunneling and therefore are proportional to square of the tunneling matrix element (and therefore, smaller compared to the off-diagonal processes described by $H_1$ and $H_2$). For triangular TQD however, each dot is connected to its two tunnel partners and therefore there is no additional smallness in $H_3$ tunneling compared to $H_1$ and $H_2$.

Summarizing, the five gates acting on the TQD (three for the quantum wells and two for the barriers) create eight random Gaussian fields (two single component random field for diagonal processes and three two-component random fields for the off-diagonal processes). The uniform noise gate acting simultaneously on all levels (not shown on the Fig. 5) can be added straightforwardly by an extra random field component.

The single electron Green's function in a particular realization of off-diagonal disorder given by the Hamiltonians $H_1$ and $H_2$ is:

$$G_r^R(\epsilon) = G_l^R(\epsilon) = \frac{(\epsilon - \epsilon_0)^2 - \frac{1}{2}\left[|\Delta_{12}|^2 + |\Delta_{34}|^2 + 2\text{Re}\left\{\Delta_{12}^*\Delta_{34}\right\}\right]}{(\epsilon - \epsilon_0 + i\delta)\left\{(\epsilon - \epsilon_0 + i\delta)^2 - \left[|\Delta_{12}|^2 + |\Delta_{34}|^2\right]\right\}}, \tag{26}$$

$$G_c^R(\epsilon) = \frac{\epsilon - \epsilon_0}{(\epsilon - \epsilon_0 + i\delta)^2 - |\Delta_{12}|^2 - |\Delta_{34}|^2}. \tag{27}$$

---

[3]The uniform random potential identically acting on all levels can be straightforwardly added, as explained in the previous subsection.

The surfaces $|\Delta_{12}|^2 + |\Delta_{34}|^2 = \Delta_1^2 + \Delta_2^2 + \Delta_3^2 + \Delta_4^2 = const$ define the spheres $S_3$ in $d = 4$ parametric space.

The most general form of the single-particle Hamiltonian is:

$$H = \left[ \epsilon_0 \cdot \hat{\mathbb{1}} + \vec{h} \cdot \hat{\vec{M}} \right]_{\alpha\beta} c_\alpha^\dagger c_\beta \,, \tag{28}$$

where $\vec{h}$ is the eight component Gaussian random potential (synthetic magnetic field) and $\hat{\vec{M}}$ is the eight-component vector consisting of eight Gell-Mann matrices.

These arguments can be repeated for the serial complex Quantum dots consisting of $M$ gated quantum wells separated by $M-1$ fluctuating barriers. Using the basis of $SU(M)$ group one straightforwardly obtains the Hamiltonian describing a particle in $M^2-1$-component random potential.

In *Études* we analyse the disorder-averaged Green's function $\bar{G}(\epsilon)$ for Keldysh models with a multi-component Gaussian random potential. Since all models discussed in this manuscript are $0+1$ dimensional with zero stands for spatial dimensions, DOS $\nu(\epsilon)$ is related to the single particle Green's functions as follows[4]:

$$\nu(\epsilon) = -\frac{1}{\pi} \mathrm{Im}\bar{G}^R(\epsilon) \,. \tag{29}$$

The single particle T-matrix is expressed in terms of the self-energy $\Sigma(\epsilon) = G_0^{-1} - \bar{G}^{-1}(\epsilon)$:

$$T(\epsilon) = G_0^{-1}(\epsilon)\Sigma(\epsilon)\bar{G}(\epsilon) = \epsilon^2 \bar{G}(\epsilon) - \epsilon \,. \tag{30}$$

Knowing the exact form of $\bar{G}^R$ gives access to all physical single particle characteristics of QDs.

## 3  *Étude* No 1. Keldysh model with one-component noise

We start our first *Étude* by considering the original Keldysh model in the time domain [6]. The model describes, for example, single electron states in a single well potential (Single Quantum Dot) under the action of a fluctuating back gate (see the the *Intermezzo*). The "scalar" single component classical Gaussian random field (noise) is associated with the slow fluctuation of the confining potential. For the purposes of our notes we assume that the characteristic time associated with the change of the confining potential is greater than any other characteristic time of the system (e.g. Zener time, dwell time etc). Since we are dealing with single particle physics, statistics do not play any role. We will consider Fermi and Bose many-body models in a separate publication. One of the goals of our *Études* is to demonstrate the properties of the model and the response functions. To achieve this we will allow ourselves to show sketches of derivation and providing some minimal details of our calculations.

The single particle Green's function (GF) is defined as follows[5]:

$$G_1^{-1}(\epsilon) = \epsilon - \Sigma_1(\epsilon) = G_0^{-1}(\epsilon) - \Sigma_1(\epsilon) \,. \tag{31}$$

Here the index (subscript) "1" refers to the one-component model. The inverse bare GF is $G_0^{-1}(\epsilon) = \epsilon$ and $\Sigma_1(\epsilon)$ is the self-energy part. As already mentioned, the classical Gaussian noise correlator is

$$D_1(\omega) = 2\pi W^2 \delta(\omega) \,, \tag{32}$$

---

[4]We omit the degeneracy associated with the spin assuming for example existence of a strong polarizing magnetic field which lifts out such degeneracy.

[5]To simplify our notations we omit *bar* in the notations of the Green's function. Everywhere below we denote by $G$ the disorder averaged GF.

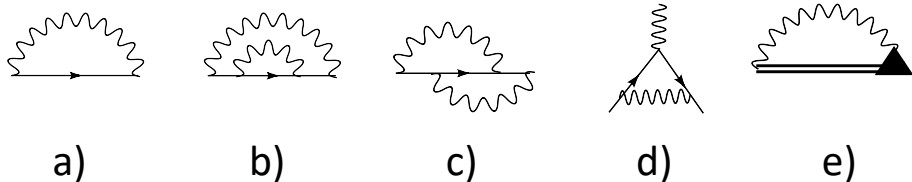

Figure 6: Self-energy part diagrams a) first order perturbation theory; b) and c) second order of perturbation theory; d) first order vertex correction; e) bold self-energy part. Single straight line denoted bare Green's function, double solid line denotes bold (dressed) Green's function. Wavy line stands for the Gaussian classical random field correlator. Bold triangle denotes the full vertex $\Gamma$.

is quasi-static as the characteristic noise time $t_{\text{noise}} \to \infty$. Parameter $W^2$ is the variance of the Gaussian distribution. The self-energy (SE) part is defined by the following equation

$$\Sigma_1(\epsilon) = \epsilon - G_1^{-1}(\epsilon) = W^2 G_1(\epsilon)\Gamma_1(\epsilon). \tag{33}$$

The first and second order Feynman diagrams for the self-energy are shown on Figure 6 a-c. SE is connected to the "triangular" vertex $\Gamma_1$ (Fig. 6 d-e).

The Ward Identity (WI)

$$\Gamma_1(\epsilon) = \frac{dG_1^{-1}(\epsilon)}{d\epsilon} = 1 - \frac{d\Sigma_1(\epsilon)}{d\epsilon} \tag{34}$$

establishes an explicit relation between $\Gamma_1$ and $\Sigma_1$ as the "incoming" through the wavy line frequency is zero.

Combining the Dyson equation (31 - 33) with the Ward Identity (34)

$$W^2 \frac{dG_1(\epsilon)}{d\epsilon} + \epsilon G_1(\epsilon) = 1, \tag{35}$$

$$W^2 \frac{d\Sigma_1(\epsilon)}{d\epsilon} + \epsilon \Sigma_1(\epsilon) - \Sigma_1^2(\epsilon) = W^2, \tag{36}$$

we get a closed set of linear first order ordinary differential equations for GF (35) and a non-linear first order Riccati-type differential equation (36) for SE.[6]

Boundary condition

$$G_1(\epsilon \to \infty) = \frac{1}{\epsilon} \tag{37}$$

determines the large frequency behaviour of GF.

It is convenient to introduce the following dimensionless variables and functions:

$$G_1(\epsilon) = \frac{1}{W}\Psi_1\left(z = \frac{\epsilon}{W}\right), \quad \Sigma_1(\epsilon) = W\sigma_1\left(z = \frac{\epsilon}{W}\right). \tag{38}$$

As a result, the dimensionless equations for GF and SE

$$\boxed{\frac{d\Psi_1(z)}{dz} + z\Psi_1(z) = 1,} \tag{39}$$

---

[6] Note that both equations originate from the Dyson equation. The equation for $\Sigma_1$ does not contain any new information. Therefore, it is sufficient to solve just one equation (e.g. for $G_1$) and use a non-linear connection between $G_1$ and $\Sigma_1$ to obtain the self-energy part. However in this $\acute{E}tude$ we show how to solve each equation to demonstrate the existence of a non-trivial differential identity between GF and SE.

$$\boxed{\frac{d\sigma_1(z)}{dz} + z\sigma_1(z) - \sigma_1^2(z) = 1} \tag{40}$$

are accompanied by the boundary conditions:

$$\Psi_1(z \to \infty) = \frac{1}{z}, \qquad \sigma_1(z \to \infty) = \frac{1}{z}. \tag{41}$$

The non-linear Riccati equation (40) can be transferred to a linear second order ordinary differential equation by substitution

$$\sigma_1(z) = -\frac{1}{\Phi_1(z)} \frac{d\Phi_1(z)}{dz} = -\frac{d}{dz} \log[\Phi_1(z)]. \tag{42}$$

Corresponding boundary condition for the new function $\Phi_1(z)$ is

$$\Phi_1(z \to \infty) = \frac{1}{z}. \tag{43}$$

The equation for $\Phi_1(z)$ reads as follows:

$$\frac{d^2\Phi_1(z)}{dz^2} + z\frac{d\Phi_1(z)}{dz} + \Phi_1(z) = 0, \tag{44}$$

or, equivalently,

$$\frac{d}{dz}\left[\frac{d\Phi_1(z)}{dz} + z\Phi_1(z)\right] = 0. \tag{45}$$

Using the boundary condition Eq.(43) we re-write the equations in the following form:

$$\frac{d\Phi_1(z)}{dz} + z\Phi_1(z) = 1, \tag{46}$$

and therefore $\Phi_1(z) \equiv \Psi_1(z)$. Finally, the Ward Identity casts the form

$$\sigma_1(z) = -\frac{d}{dz}\log[\Psi_1(z)], \quad \Gamma_1(z) = 1 - \frac{d\sigma_1(z)}{dz} = 1 + \frac{d^2}{dz^2}\log[\Psi_1(z)], \tag{47}$$

or, equivalently,

$$\sigma_1(z) = z - \frac{1}{\Psi_1(z)}, \qquad \Gamma_1(z) = \frac{\sigma_1(z)}{\Psi_1(z)} = \frac{z\Psi_1(z) - 1}{\Psi_1^2(z)}. \tag{48}$$

Obviously, these equations can be obtained directly from the definition of $\sigma_1(z)$, the Ward identity and Eq.(39).

**Analytic properties**

This subsection is devoted to a discussion of the analytic properties of the single particle Green's function. The retarded dimensionless Green's function $\Psi_1^R(z)$ can be obtained with the help of the Weierstrass transformation:

$$\Psi_1^R(z) = \frac{1}{\sqrt{2\pi}} \int_{-\infty}^{\infty} du\, \psi_1^R(u - z) e^{-u^2/2}. \tag{49}$$

Substituting it to the Dyson equation we get:

$$\Psi_1^R(z) = \frac{1}{\sqrt{2\pi}} \int_{-\infty}^{\infty} du\, \frac{e^{-u^2/2}}{z - u + i\delta}. \tag{50}$$

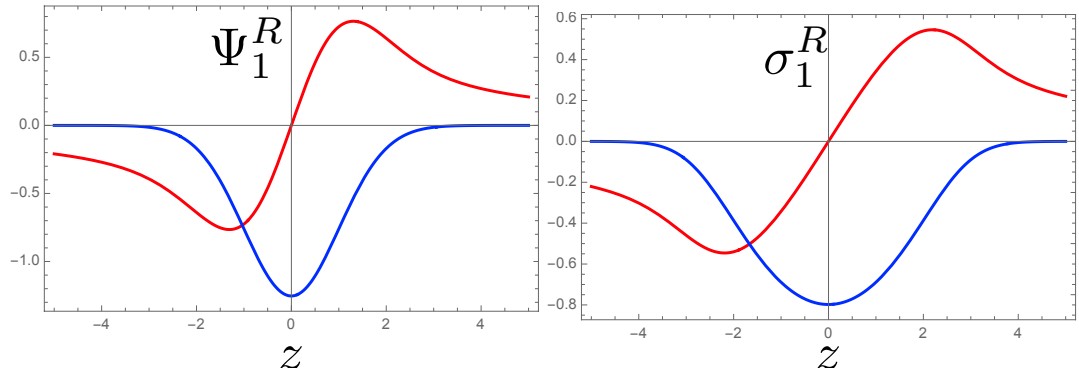

Figure 7: Left: $\text{Re}[\Psi_1^R(z)]$ (red curve), $\text{Im}[\Psi_1^R(z)]$ (blue curve). Right: $\text{Re}[\sigma_1^R(z)]$ (red curve), $\text{Im}[\sigma_1^R(z)]$ (blue curve).

The imaginary part $\text{Im}\Psi_1^R(z)$ can be easily computed:

$$\boxed{\text{Im}\Psi_1^R(z) = -\sqrt{\frac{\pi}{2}}e^{-z^2/2} \xrightarrow{z\to 0} -\sqrt{\frac{\pi}{2}}\left[1 - \frac{z^2}{2} + O(z^4)\right].} \tag{51}$$

The real part $\text{Re}\Psi_1^R(z)$ can be determined by the Kramers - Kronig relations

$$\text{Re}\Psi_1^R(z) = \frac{1}{\pi}\text{P.V.}\int_{-\infty}^{\infty} du \frac{\text{Im}\Psi_1^R(u)}{u - z}. \tag{52}$$

Here P.V. denotes Cauchy principle value of the integral. The above transformation is connected to the Hilbert transform (HT) of the Gaussian function $H[e^{-u^2}](z) = H_0(z)$ [10] as shown in Appendix B:

$$H_0(z) = \frac{1}{\pi}\text{P.V.}\int_{-\infty}^{\infty} \frac{e^{-u^2}}{z - u}du. \tag{53}$$

Evaluating Eq. (53) at $z = 0$ we obtain the obvious property $\text{Re}\Psi_1^R(z = 0) = 0$ which can be used as a new initial condition for the linear ordinary differential equation for the function $\psi_1(z) = \text{Re}\Psi_1^R(z)$:

$$\frac{d\psi_1}{dz} + z\psi_1 = 1, \quad \psi_1(0) = 0. \tag{54}$$

The solution of this equation is obtained by the method of indefinite coefficients and finally reads as:

$$\psi_1(z) = C \cdot e^{-z^2/2} + e^{-z^2/2}\int_0^z e^{u^2/2}du. \tag{55}$$

Implementing the initial condition $\psi_1(0) = 0$ we obtain $C = 0$. Finally, $\text{Re}\Psi_1^R(z)$ can be expressed in terms of the Dawson function $D_+(z) = F(z)$ [11]:

$$\text{Re}\Psi_1^R(z) = \sqrt{2}F\left(\frac{z}{\sqrt{2}}\right), \quad F(z) = \frac{\sqrt{\pi}}{2}e^{-z^2}\text{erfi}(z). \tag{56}$$

One can represent the Dawson function as the sin- Fourier-Laplace transform of the Gaussian function:

$$F(z) = \frac{1}{2}\int_0^{\infty} ds\, e^{-s^2/4}\sin zs. \tag{57}$$

The Dawson function can be expanded as follows for small values of the argument $z \to 0$:

$$F(z \to 0) = \sum_{n=0}^{\infty} \frac{(-1)^n 2^n}{(2n+1)!!} z^{2n+1}, \tag{58}$$

and in the limit $z \to \infty$ the expansion reads:

$$F(z \to \infty) = \sum_{n=0}^{\infty} \frac{(2n-1)!!}{2^{n+1} z^{2n+1}}. \tag{59}$$

Keeping first few terms of the Dawson function for small/large values of its argument, we get:

$$\mathrm{Re}\Psi_1^R(z) = \sqrt{2} F\left(\frac{z}{\sqrt{2}}\right) \xrightarrow{z \to 0} \left[z - \frac{1}{3}z^3 + O\left(z^5\right)\right] \tag{60}$$

$$\xrightarrow{z \to \infty} \left[\frac{1}{z} + \frac{1}{z^3} + O\left(\frac{1}{z^5}\right)\right]. \tag{61}$$

Obviously, since GF is purely imaginary in the zero frequency limit $\Psi_1^R(0) = -i\sqrt{\frac{\pi}{2}}$, the modulus square is $|\Psi_1^R(0)|^2 = \frac{\pi}{2}$. We plot the frequency dependence of the $\mathrm{Re}\Psi_1$ and $\mathrm{Im}\Psi_1$ in the left panel of Fig.7.[7]

Substituting the Taylor expansion for $\Psi_1^R(z)$ into the equation for the self-energy

$$\sigma_1^R(z) = z - \frac{\mathrm{Re}\Psi_1^R(z) - i\mathrm{Im}\Psi_1^R(z)}{|\Psi_1^R(z)|^2}, \tag{64}$$

we obtain the asymptotic behaviour of the self-energy at $z \to 0$

$$\mathrm{Re}\sigma_1^R(z) \xrightarrow{z \to 0} \left(1 - \frac{2}{\pi}\right)z + \frac{4(\pi-3)z^3}{3\pi^2} + O\left(z^4\right), \tag{65}$$

$$\mathrm{Im}\sigma_1^R(z) \xrightarrow{z \to 0} -\sqrt{\frac{2}{\pi}} - \frac{(\pi-4)z^2}{\sqrt{2}\pi^{3/2}} + O\left(z^4\right). \tag{66}$$

In the $z \to \infty$ limit

$$\mathrm{Re}\sigma_1^R(z) \xrightarrow{z \to \infty} \frac{1}{z} + \frac{2}{z^3} + O\left(\frac{1}{z^5}\right). \tag{67}$$

The frequency dependencies of $\mathrm{Re}\sigma_1$ and $\mathrm{Im}\sigma_1$ are shown in the right panel of Fig.7. The DOS for the single component Keldysh model is represented by a single Gaussian-shaped peak centred at $z = 0$.

---

[7]The real-time GF (dimensionless time $s = t \cdot W$) is defined through the Fourier transformation:

$$\Psi_1^R(s) = \int_{-\infty}^{\infty} \frac{dz}{2\pi} \Psi_1^R(z) e^{-izs} = -i e^{-s^2/2} \Theta(s). \tag{62}$$

Here $\Theta(s)$ is a step (Heaviside) function. The real time GF satisfies the differential equation

$$\frac{d\Psi_1^R(s)}{ds} + s \cdot \Psi_1^R(s) = -i\delta(s). \tag{63}$$

One can notice a duality of Dyson equations in the time and frequency domain. This duality is associated with the Gaussian form of the random field.

# 4 *Étude* No 2. Keldysh model with two-component noise

The two-component time-dependent Keldysh model describes, for example, the single electron states in a double well potential under action of slow fluctuations of the barrier separating the wells (see the *Intermezzo*) [6]. The complex tunnel matrix element has two independent components representing two Gaussian random fields (noise) in the x- and y- pseudospin directions.

The single particle Green's function of the two-component time dependent Keldysh model is defined as follows

$$G_2^{-1}(\epsilon) = \epsilon - \Sigma_2(\epsilon) = G_0^{-1}(\epsilon) - \Sigma_2(\epsilon), \tag{68}$$

where $G_0^{-1}(\epsilon) = \epsilon$ is the inverse barer GF. The two-component Gaussian random field correlator

$$D_2(\omega) = 2 \times 2\pi W^2 \delta(\omega) \tag{69}$$

describes the two-component slow noise. For simplicity, we assume that two independent amplitudes of the noise coincide. Feynman diagrams for the Green's function before Gaussian averaging are shown on Fig. 8. At each vertex of the diagram the "pseudo-spin" is flipped and therefore black and white vertices form the staggered pattern. Dashed lines denote the Gaussian random potential.

The self-energy part is proportional to the product of the bold GF $G_2$ and the triangular vertex $\Gamma_2$

$$\Sigma_2(\epsilon) = \epsilon - G_2^{-1}(\epsilon) = W^2 G_2(\epsilon)\Gamma_2(\epsilon). \tag{70}$$

Several first irreducible diagrams for the self-energy part are shown in Fig. 9. Each "black" vertex is connected only to a "white" vertex and vice-versa. As a result, there exists no second order irreducible diagram.[8]

Before we derive the differential equation for $G_2$ we show that GF can be obtained by direct re-summation of the Feynman diagrams of the "formal" perturbation theory

$$G_2^R(\epsilon) = G_0(\epsilon) + \sum_{n=1}^{\infty} A_n^{(2)}(W)^{2n} G_0^{2n+1}(\epsilon), \tag{71}$$

where the coefficients $A_n^{(2)} = (2n)!!$ reflect the fact that all diagrams at a given order of the perturbation theory have the same value. To proceed with the re-summation we use the integral representation for the Euler's Gamma-function. As the next step, we compute explicitly the sum over $n$ to transform the integral to the form

$$G_2^R(\epsilon) = \int_0^{\infty} 2t\,dt \frac{G_0(\epsilon)}{1 - 2t^2 W^2 G_0^2(\epsilon)} e^{-t^2}. \tag{72}$$

As a result, the Green's function is expressed in terms of the double integrals:

$$G_2^R(\epsilon) = \frac{1}{2} \int_{-\infty}^{+\infty} \int_{-\infty}^{+\infty} \frac{dx\,dy\, e^{-(x^2+y^2)/2W^2}}{2\pi W^2} \left[ \frac{1}{\epsilon - \sqrt{x^2 + y^2} + i\delta} + \frac{1}{\epsilon + \sqrt{x^2 + y^2} + i\delta} \right]. \tag{73}$$

Transforming this integral from the "Cartesian" coordinates to the "polar" coordinate system one gets:

$$G_2^R(\epsilon) = \int_0^{\infty} \frac{u\,du}{2W^2} \left( \frac{1}{\epsilon - u + i\delta} + \frac{1}{\epsilon + u + i\delta} \right) e^{-u^2/2W^2}. \tag{74}$$

---

[8]The reducible second order diagram and other reducible diagrams in all orders result in replacement of the bare GF in diagram Fig. 9 to bold GF.

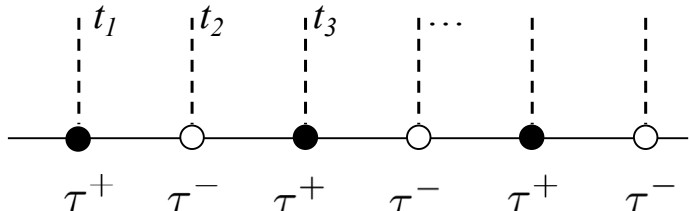

Figure 8: Feynman diagram for the GF before taking averaging over two-component Gaussian random field.

Due to the $SO(2)$ symmetry of the noise the integral is independent on the polar angle. Here we would like to note that in fact the Eq. (74) describes an averaging of GF with the known Rayleigh distribution. Finally, by differentiating Eq. (74) with respect to the frequency $\epsilon$ we obtain a linear differential equation for the single particle Green's function.[9]

$$W^2 \frac{dG_2(\epsilon)}{d\epsilon} + \epsilon G_2(\epsilon)\left(1 - \frac{W^2}{\epsilon^2}\right) = 1 \,. \tag{75}$$

The same equation can be derived with the help of the Ward identity

$$\Gamma_2(\epsilon) = \frac{1}{\epsilon}\frac{d}{d\epsilon}\left[\epsilon G_2^{-1}(\epsilon)\right] \,. \tag{76}$$

The corresponding Dyson equation for the self-energy functions reads as follows:

$$W^2 \frac{d\Sigma_2(\epsilon)}{d\epsilon} + W\left[\frac{\epsilon}{W} + \frac{W}{\epsilon}\right]\Sigma_2(\epsilon) - \Sigma_2^2(\epsilon) = 2W^2 \,. \tag{77}$$

The boundary condition reads:

$$G_2(\epsilon \to \infty) = \frac{1}{\epsilon} \,, \qquad \Sigma_2(\epsilon \to \infty) = \frac{2}{\epsilon} \,. \tag{78}$$

Similarly to the first $\acute{E}tude$ we introduce dimensionless variables and functions:

$$G_2(\epsilon) = \frac{1}{W}\Psi_2\left(z = \frac{\epsilon}{W}\right) \,, \qquad \Sigma_2(\epsilon) = W\sigma_2\left(z = \frac{\epsilon}{W}\right) \,. \tag{79}$$

As a result, the dimensionless equations cast the form:

$$\boxed{\frac{d\Psi_2(z)}{dz} + \left[z - \frac{1}{z}\right]\Psi_2(z) = 1 \,,} \tag{80}$$

$$\boxed{\frac{d\sigma_2(z)}{dz} + \left[z + \frac{1}{z}\right]\sigma_2(z) - \sigma_2^2(z) = 2 \,,} \tag{81}$$

being accompanied by the corresponding boundary conditions:

$$\Psi_2(z \to \infty) = \frac{1}{z} \,, \qquad \sigma_2(z \to \infty) = \frac{2}{z} \,. \tag{82}$$

It can be seen that in contrast the the one-component Keldysh model, an additional $1/z$ "centrifugal" term appears in the Dyson equation. As we will see below, this term is responsible for level repulsion.

---

[9]To simplify our notations we omit below the index $R$ (retarded) in the superscript of the GF. Basically, it means that we will be dealing with the real part of GF. We restore the index $R$ in the next subsection when considering analytic properties of GF and SE.

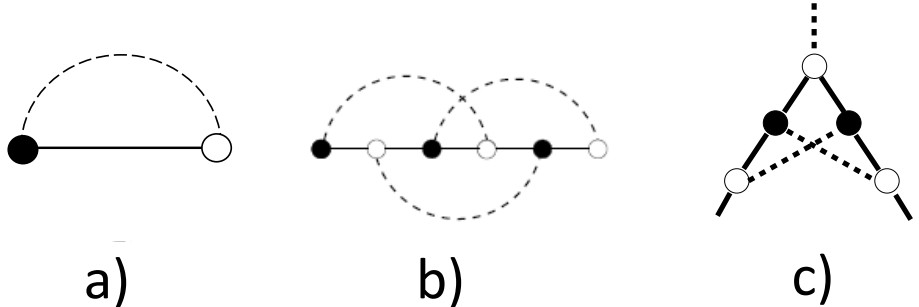

Figure 9: Irreducible self-energy part diagram in the first a) and third b) orders of the perturbation theory. Second order diagram is topological impossible. c) first non-trivial diagram for the vertex.

The boundary condition for $\sigma_2$ follows from the Ward Identity and the boundary condition for $\Psi_2(z)$:

$$\Gamma_2(z)|_{z \to \infty} = \frac{1}{z} \frac{d}{dz} \left[ z \Psi_2^{-1}(z) \right] = \frac{\Psi_2^{-1}(z)}{z} + \frac{d\Psi_2^{-1}(z)}{dz} = 2. \tag{83}$$

Therefore we have

$$\sigma_2(z) = \Psi_2(z) \Gamma_2(z) \xrightarrow{z \to \infty} \frac{2}{z}. \tag{84}$$

The solution of Eq. (80) with the boundary condition Eq. (82) is given by

$$\Psi_2(z) = \frac{1}{2} \int_0^\infty u e^{-u^2/2} du \left[ \frac{1}{z-u} + \frac{1}{z+u} \right]. \tag{85}$$

The non-linear equation (81) for the SE is a Riccati-type equation. Substituting

$$\sigma_2(z) = -\frac{1}{\Phi_2(z)} \frac{d\Phi_2(z)}{dz} = -\frac{d}{dz} \log \left[ \Phi_2(z) \right], \tag{86}$$

and taking into account the boundary condition Eq. (82) we get the corresponding boundary condition for the auxiliary function $\Phi_2$

$$\Phi_2(z \to \infty) = \frac{1}{z^2}. \tag{87}$$

The second-order linear ordinary differential equation (ODE) for $\Phi_2(z)$ reads as follows:

$$\frac{d^2\Phi_2(z)}{dz^2} + \left[ z + \frac{1}{z} \right] \frac{d\Phi_2(z)}{dz} + 2\Phi_2(z) = 0, \tag{88}$$

which can be re-written in more compact form:

$$\frac{d}{dz} R(z) + \frac{1}{z} R(z) = 0. \tag{89}$$

Here we introduced a new function

$$R(z) = \frac{d\Phi_2(z)}{dz} + z\Phi_2(z). \tag{90}$$

The corresponding boundary condition for $R(z)$ reads as:

$$R(z \to \infty) = \frac{1}{z}. \tag{91}$$

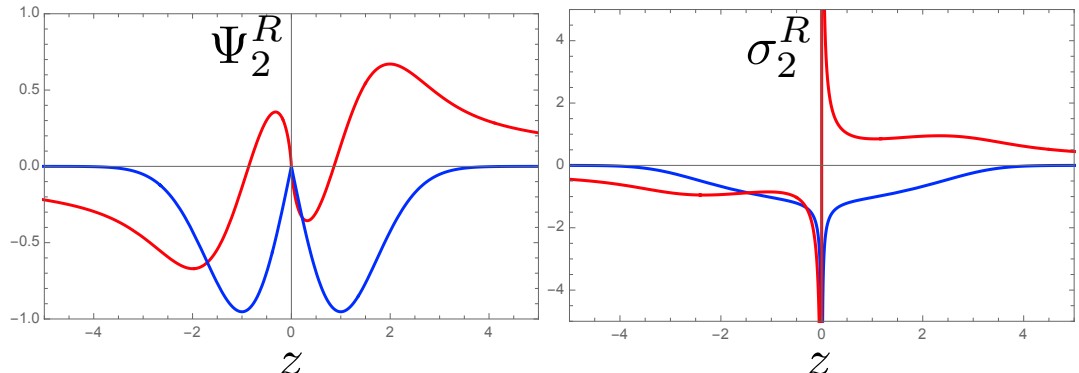

Figure 10: Left: $\text{Re}[\Psi_2^R(z)]$ (red curve), $\text{Im}[\Psi_2^R(z)]$ (blue curve). Right: $\text{Re}[\sigma_2^R(z)]$ (red curve), $\text{Im}[\sigma_2^R(z)]$ (blue curve).

The general solution of the equation (89) is $R(z) = A/z$ and the boundary condition (91) defines the constant $A = 1$.

Now we use the definition of $R(z)$ to obtain a first order linear ODE for $\Phi_2(z)$:

$$\frac{d\Phi_2(z)}{dz} + z\Phi_2(z) = \frac{1}{z}. \tag{92}$$

Substituting $\Phi_2(z) = \frac{\Theta_2(z)}{z}$ we obtain the ODE for $\Theta_2$:

$$\frac{d\Theta_2(z)}{dz} + \left[z - \frac{1}{z}\right]\Theta_2(z) = 1, \tag{93}$$

with the boundary condition

$$\Theta_2(z \to \infty) = \frac{1}{z}. \tag{94}$$

By comparing Eq. (93) with Eq. (80) we conclude that $\Theta_2(z) \equiv \Psi_2(z)$. In addition we obtain new differential identities:

$$\sigma_2(z) = -\frac{d}{dz}\log\left[\frac{\Psi_2(z)}{z}\right], \quad \Gamma_2(z) = 1 - \frac{d\sigma_2(z)}{dz} = 1 + \frac{d^2}{dz^2}\log\left[\frac{\Psi_2(z)}{z}\right]. \tag{95}$$

The connection between the functions $\sigma_2(z)$, $\Gamma_2(z)$ and $\Psi_2(z)$ is given by the equations identical to Eq. (48):

$$\sigma_2(z) = \frac{z\Psi_2(z) - 1}{\Psi_2(z)}, \quad \Gamma_2(z) = \frac{z\Psi_2(z) - 1}{\Psi_2^2(z)}. \tag{96}$$

**Analytic properties**

The retarded dimensionless Green's function $\Psi_2^R(z)$ is given by

$$\Psi_2^R(z) = \frac{1}{2}\int_0^\infty u e^{-u^2/2} du\left[\frac{1}{z - u + i\delta} + \frac{1}{z + u + i\delta}\right]. \tag{97}$$

The imaginary part $\text{Im}\Psi_2^R(z)$ can be easily computed as:

$$\text{Im}\Psi_2^R(z) = -\frac{\pi}{2}z\,\text{sgn}\,z\,e^{-z^2/2} \xrightarrow{z\to 0} -\frac{\pi}{2}z\,\text{sgn}\,z\left[1 - \frac{z^2}{2} + O(z^4)\right]. \tag{98}$$

The identity $\mathrm{Re}\Psi_2^R(z=0)=0$ follows directly from the analytic properties of $\Psi_2$. The function $\psi_2(z)=\mathrm{Re}\Psi_2^R(z)$ satisfies the ordinary linear differential equation:

$$\frac{d\psi_2}{dz}+\left[z-\frac{1}{z}\right]\psi_2=1\,,\quad \psi_2(0)=0\,. \tag{99}$$

The homogeneous solution is given by:

$$\psi_2^{(h)}=C\cdot z\,e^{-z^2/2}\,. \tag{100}$$

The particular solution is obtained by the method of indefinite coefficients which leads to another differential equation for the function $C(z)$:

$$\frac{dC(z)}{dz}=\frac{e^{z^2/2}}{z}\,. \tag{101}$$

Finally, the solution of the boundary problem is given by:

$$\psi_2(z)=\frac{1}{2}z\,e^{-z^2/2}\mathrm{Ei}\left[z^2/2\right]\,. \tag{102}$$

The real part $\mathrm{Re}\Psi_2^R(z)$ is expressed through the Hilbert transform $H_1[ue^{-u^2}]$ (see details in Appendix B):

$$\mathrm{Re}[\Psi_2^R(z)]=\frac{1}{2}\mathrm{P.V.}\int_0^\infty ue^{-u^2/2}du\left[\frac{1}{z-u}+\frac{1}{z+u}\right]=\frac{1}{2}z\,e^{-z^2/2}\mathrm{Ei}\left[z^2/2\right] \tag{103}$$

$$\xrightarrow{z\to 0} z\log|z|+\frac{1}{2}\left[\gamma-\log 2\right]z+O(z^3)\,. \tag{104}$$

Here $\mathrm{Ei}(u)$ is the exponential integral function, $\gamma$ is the Euler constant, $|\Psi_2(z\to 0)|^2\propto z^2\log^2[z]$. Note, the $\mathrm{Im}\Psi_2^R(z\to 0)\propto |z|$ is not differentiable at $z=0$. Furthermore, $\mathrm{Re}\Psi_2^R(z\to 0)\propto z\log|z|$ is not analytic in the limit $z\to 0$ and have a branch cut. So GF cannot be decompose into a Taylor expansion in the limit $z\to 0$. The frequency dependencies of the real and imaginary parts of GF $\Psi_2$ are shown in the left panel of Fig. 10.[10]

The self-energy is diverging and non-analytic in the limit $z\to 0$:

$$\mathrm{Re}\sigma_2^R(z)\xrightarrow{z\to 0}-\frac{1}{z\log|z|}\,, \tag{106}$$

$$\mathrm{Im}\sigma_2^R(z)\xrightarrow{z\to 0}\frac{\pi}{2|z|\log^2|z|}\,. \tag{107}$$

The asymptotic behaviour of GF in the limit $z\to\infty$ is:

$$\mathrm{Re}[\Psi_2^R(z)]\xrightarrow{z\to\infty}\frac{1}{z}+\frac{2}{z^3}+O\left(\frac{1}{z^5}\right)\,, \tag{108}$$

and $\sigma_2(z\to\infty)\to 2/z+O(1/z^3)$. The frequency dependencies of the real and imaginary parts of the self-energy $\sigma_2$ are shown on the right panel of Fig. 10.

The DOS for the two-component Keldysh model is represented by two Gaussian peaks separated by a pseudo-gap with linear energy dependence in the vicinity $z=0$.

---

[10]Note that the real-time GF obtained by sin - Fourier transform can be written in terms of the Dawson function:

$$\psi_2(s)=\int_{-\infty}^\infty dz\,\psi_2(z)\sin zs=\pi\left(1-s\sqrt{2}F\left(\frac{s}{2}\right)\right)\,. \tag{105}$$

# 5 *Étude* No 3. Keldysh model with three-component noise

We start this *Étude* by considering a Double Quantum Dot realization of the Keldysh model. We assume that there exists a random Gaussian field proportional to $\tau_x$, $\tau_y$ Pauli matrices, which is associated with the barrier fluctuations, and an additional *alternating* diagonal random Gaussian field acting on the levels in each dot. The last field is proportional to $\tau_z$. A realization of the alternating diagonal noise is done by applying two separate back gates acting to each QD. All three generators of SU(2) group are contained in the Hamiltonian. Without losing generality, we consider the case of equal noise amplitudes similarly to previous *Étude*. Breaking the rotational symmetry is discussed in *Variation No.* 1. Another realization of the three component random field can be implemented when barrier fluctuations are accompanied by a *uniform* random Gaussian field acting on the levels of each QD through a single back gate applied to both QD in the double well potential. This noise is proportional to the unit matrix $\tau_0$. We discuss it in *Variation No.* 2.

The general equations relating the dimensionless SE $\sigma_3(z)$ and the dimensionless triangular vertex $\Gamma_3(z)$ with the single particle GF $\Psi_3(z)$ are:

$$\sigma_3(z) = \frac{z\Psi_3(z)-1}{\Psi_3(z)}, \qquad \Gamma_3(z) = \frac{z\Psi_3(z)-1}{\Psi_3^2(z)}. \tag{109}$$

These equations are obtained directly from the definition of $\sigma_3(z)$ and the Ward Identity

$$\boxed{\sigma_3(z) = \Psi_3(z)\Gamma_3(z), \qquad \Gamma_3(z) = \frac{1}{z^2}\frac{d}{dz}\left[z^2\Psi_3^{-1}(z)\right].} \tag{110}$$

The differential equation for the $\Psi_3(z)$ and $\sigma_3(z)$ and the boundary conditions are given by

$$\boxed{\frac{d\Psi_3(z)}{dz} + \left[z - \frac{2}{z}\right]\Psi_3(z) = 1, \quad \Psi_3(z\to\infty) = \frac{1}{z},} \tag{111}$$

$$\boxed{\frac{d\sigma_2(z)}{dz} + \left[z + \frac{2}{z}\right]\sigma_2(z) - \sigma_2^2(z) = 3.} \tag{112}$$

The solution of the linear ODE for $\Psi_3(z)$ casts the form:

$$\Psi_3(z) = \frac{1}{2\Omega_3}\int_0^\infty u^2 e^{-u^2/2}du\left[\frac{1}{z-u} + \frac{1}{z+u}\right]. \tag{113}$$

Here $\Omega_3 = \sqrt{\pi/2}$ is the normalizing coefficient. Equivalently, the solution can be written in terms of the Hilbert transform $H[u^2 e^{-u^2/2}](z)$ (see Appendix B) as:

$$\Psi_3(z) = \frac{1}{2\Omega_3}\int_{-\infty}^\infty du\frac{u^2 e^{-u^2/2}}{z-u}. \tag{114}$$

Less trivial identities

$$\boxed{\sigma_3(z) = -\frac{d}{dz}\log\left[\frac{\Psi_3(z)}{z^2}\right], \qquad \Gamma_3(z) = 1 - \frac{d\sigma_3(z)}{dz} = 1 + \frac{d^2}{dz^2}\log\left[\frac{\Psi_3(z)}{z^2}\right].} \tag{115}$$

are obtained by solving a non-linear Riccati-type equation for $\sigma_3(z)$.

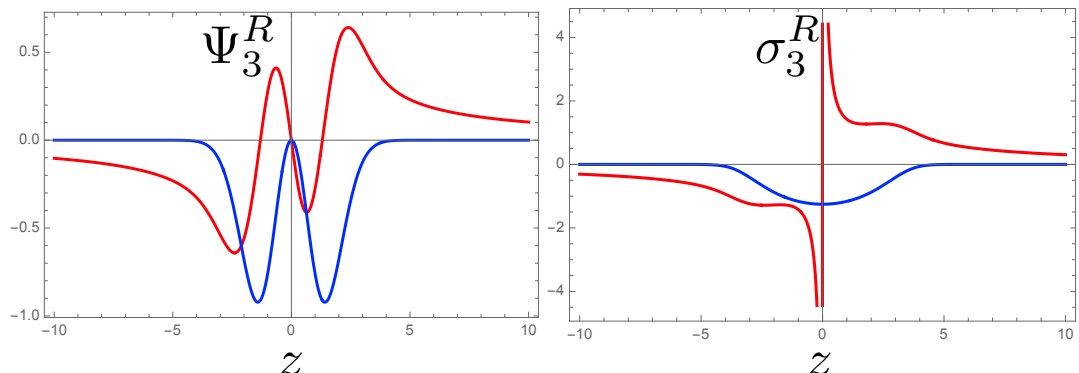

Figure 11: Left: $\mathrm{Re}[\Psi_3^R(z)]$ (red curve), $\mathrm{Im}[\Psi_3^R(z)]$ (blue curve). Right: $\mathrm{Re}[\sigma_3^R(z)]$ (red curve), $\mathrm{Im}[\sigma_3^R(z)]$ (blue curve).

**Analytic properties**

The retarded dimensionless Green's function $\Psi_3^R(z)$ is given by the equation

$$\Psi_3^R(z) = \frac{1}{\sqrt{2\pi}} \int_{-\infty}^{\infty} du \, \frac{u^2 e^{-u^2/2}}{z - u + i\delta} \,. \tag{116}$$

The imaginary part $\mathrm{Im}\Psi_3^R(z)$ is easily computed

$$\boxed{\mathrm{Im}\Psi_3^R(z) = -\sqrt{\frac{\pi}{2}} z^2 e^{-z^2/2} \xrightarrow{z \to 0} -\sqrt{\frac{\pi}{2}} \left[ z^2 - \frac{z^4}{2} + O(z^4) \right].} \tag{117}$$

The real part of the analytic in the upper half-plane of $z$ function $\mathrm{Re}\Psi_3^R(z)$ can be determined by the Kramers - Kronig relations:

$$\mathrm{Re}\Psi_3^R(z) = \frac{1}{\pi} \mathrm{P.V.} \int_{-\infty}^{\infty} du \, \frac{\mathrm{Im}\Psi_3^R(u)}{u - z} \,. \tag{118}$$

It can be also expressed by the Hilbert transform $H[u^2 e^{-u^2}](z) = H_2(z)$ [10] (see Appendix B) which, in turn, is re-written in terms of the Dawson function as follows:

$$H_2(z) = -\frac{1}{\sqrt{\pi}} z + \frac{2}{\sqrt{\pi}} z^2 F(z) \,. \tag{119}$$

Substituting $H_2(z)$ to the definition of $\mathrm{Re}\Psi_3^R(z)$ we get:

$$\mathrm{Re}\Psi_3^R(z) \quad = \quad -z + \sqrt{2} z^2 F\left(\frac{z}{\sqrt{2}}\right) \tag{120}$$

$$\xrightarrow{z \to 0} \quad -z + z^3 + O(z^5) \tag{121}$$

$$\xrightarrow{z \to \infty} \quad \frac{1}{z} + \frac{3}{z^3} + O\left(\frac{1}{z^5}\right) \,. \tag{122}$$

Using the properties of the Dawson function for small values of its argument, we obtain $|\Psi_3^R(z \to 0)|^2 \propto z^2$. The frequency dependence of the real and imaginary parts of the GF $\Psi_3$ is shown in the left panel of Fig. 11.

Substituting the Taylor expansion for $\Psi_3^R(z)$ into the equation for the self-energy

$$\sigma_3^R(z) = z - \frac{\mathrm{Re}\Psi_3^R(z) - i\mathrm{Im}\Psi_3^R(z)}{|\Psi_3^R(z)|^2} \,, \tag{123}$$

we obtain the real and imaginary parts of the self-energy:

$$\mathrm{Re}\sigma_3^R(z) \xrightarrow{z\to 0} \frac{1}{z} + \left(2 - \frac{\pi}{2}\right)z + \left(\frac{2}{3} - \pi + \frac{\pi^2}{4}\right)z^3 + O\left(z^4\right), \tag{124}$$

$$\mathrm{Im}\sigma_3^R(z) \xrightarrow{z\to 0} -\sqrt{\frac{\pi}{2}} + \frac{1}{2}(\pi - 3)\sqrt{\frac{\pi}{2}}z^2 + O\left(z^4\right). \tag{125}$$

The frequency dependencies of the real and imaginary parts of the SE $\sigma_3$ are shown on the right panel of Fig. 10.

The DOS for the three-component Keldysh model is represented by two Gaussian peaks separated by a pseudo-gap with the energy-square dependence at the vicinity $z = 0$.

### *Variation* **No. 1: Anisotropic** *two + one* **Keldysh model: Breaking rotational symmetry.**

In this *Variation* we assume that the amplitudes of the random complex Gaussian field associated with the fluctuations of the inter-dot barrier (which we denote below by $W_\perp$) and the amplitude of the random scalar Gaussian field associated with the fluctuations of the level position in each well of the double well potential, denoted by $W_\parallel$ are different. In this case it is not possible any longer to introduce the dimensionless functions of the dimensionless argument with a unique re-scaling parameter. For convenience of the reader we will not rescale the variables in this Subsection. The simplest way to calculate the Green function of the anisotropic three parametric Keldysh model is to use the path integral representation by introducing a generating functional (see details in [7]). In the extreme non-Markovian case corresponding to the "infinite memory" limit [6], the Green function of the anisotropic three-component Keldysh model (which we denote $G_{2+1}(\varepsilon)$) is expressed as follows:

$$G_{2+1}(\varepsilon) = \frac{1}{W_\parallel W_\perp^2 (2\pi)^{3/2}} \int_{-\infty}^{\infty} dz\, e^{-z^2/2W_\parallel^2} \int dw^* dw\, e^{-|w|^2/2W_\perp^2} \frac{\varepsilon}{(\varepsilon + i\delta)^2 - z^2 - |w|^2}. \tag{126}$$

Re-writing the integral in the spherical coordinate system and performing the integration over the angles we obtain after some simplification:

$$G_{2+1}(\varepsilon) = \frac{1}{2W_\perp} \int_0^{\infty} d\rho\, \rho \exp\left(-\frac{\rho^2}{2W_\perp^2}\right) \frac{\mathrm{Erf}\left(\rho\sqrt{\frac{W_\perp^2 - W_\parallel^2}{2W_\perp^2 W_\parallel^2}}\right)}{\sqrt{W_\perp^2 - W_\parallel^2}} \left(\frac{1}{\varepsilon - \rho + i\delta} + \frac{1}{\varepsilon + \rho + i\delta}\right). \tag{127}$$

Here $\mathrm{Erf}(z)$ is the Error function. Let us analyse a particular case of anisotropy, namely $W_\perp > W_\parallel$.[11] The constant energy surface for the three-component anisotropic Keldysh model in both cases is an ellipsoid which transforms into a sphere if $W_\perp = W_\parallel$ and isotropy is restored. Two extreme cases of the two-component Keldysh model $W_\parallel \to 0$ of the easy-plane anisotropy and one-component Keldysh model $W_\perp \to 0$ of the easy axis anisotropy can be obtained straightforwardly. To proceed with remaining integration over $\rho$ it is convenient to use the series expansion of the Error function

$$\mathrm{Erf}(z) = \frac{2}{\sqrt{\pi}} \sum_{n=0}^{\infty} \frac{(-1)^n z^{2n+1}}{n!(2n+1)}. \tag{128}$$

As a result, Eq. (127) is transformed into

$$G_{2+1}(\varepsilon) = 2\sum_{n=0}^{\infty}\sum_{m=0}^{\infty} C_{nm}\alpha^{2m}\frac{W_\perp^{2n+1}}{W_\parallel^{2m+1}}G_0^{2n+1}(\varepsilon). \tag{129}$$

---

[11]The complimentary anisotropic limit $W_\perp < W_\parallel$ can be considered in a similar manner.

Here $\alpha = \sqrt{(W_\perp^2 - W_\parallel^2)/2}$, and $G_0(\epsilon) = (\epsilon + i\delta)^{-1}$. The coefficients

$$C_{nm} = \frac{[2(m+n)+1]!!}{m!(2m+1)} \tag{130}$$

are the combinatorial coefficients in the $n$-th order of the "three-color" diagrammatic expansion for the full anisotropic three-component Keldysh model. In the isotropic limit $\alpha = 0$ and only the term $m = 0$ survives. The coefficient $C_{n0} = A_n^{(3)} = (2n+1)!!$.

*Variation* **No. 2: Combined** *one + two* **Keldysh model**

Let us assume that in addition to the slow barrier fluctuations in the double QD there exists a uniform noise to both QDs, which is produced by back gate. For simplicity we suppose that the noise amplitudes for the one-component uniform diagonal Gaussian random potential and the two-component off-diagonal Gaussian random potential are the same. The dimensionless GF $\Psi_{1+2}(z)$ is defined as:

$$\Psi_{1+2}(z) = \frac{1}{\sqrt{2\pi}} \int_{-\infty}^{\infty} dx\, e^{-(x-z)^2/2} \Psi_2(x), \tag{131}$$

which is, in fact, the Weierstrass transform of the two-component Keldysh model Green's function.

Green's function $\Psi_{1+2}(z)$ satisfies the second order ODE which can be obtained by differentiating $\Psi_{1+2}(z)$ given by Eq. (131) with respect to it's argument, integrating by part and using the differential equation for $\Psi_2(z)$ Eq. (80):

$$\boxed{\frac{2}{z^2}\frac{d^2}{dz^2}\Psi_{1+2}(z) + \frac{3}{z}\frac{d}{dz}\Psi_{1+2}(z) + \Psi_{1+2}(z) = \frac{1}{z}.} \tag{132}$$

The corresponding boundary conditions follow from fixing two coefficients in the $z \to \infty$ asymptotic:

$$\Psi_{1+2}(z \to \infty) = \frac{1}{z} + \frac{1+2}{z^3}. \tag{133}$$

In fact, these boundary conditions (133) correspond to two conditions:

$$\Psi_{1+2}(z \to \infty) = \frac{1}{z}, \quad [z \cdot \Psi_{1+2}(z) - 1]|_{z \to \infty} = \frac{3}{z^2}. \tag{134}$$

The analytic properties of $\Psi_{1+2}(z)$ are fully defined by the corresponding properties of $\Psi_2(z)$ (see previous *Étude*).

# 6  *Étude* No 4. Keldysh model with $D \gg 1$-component noise

Let us consider the Keldysh model with a $d$-color Gaussian random potential having both diagonal (longitudinal) and off-diagonal (transverse) components. The GF in a given realization of disorder (before averaging) has a pole which defines a $D = d - 1$ -dimensional sphere $S_D$ (see *Intermezzo*).

For consistency with the notations of the previous *Études*: $d = 1$ corresponds to the one-component KM, $d = 2$ describes the two-component KM on a circle $S_1$ and $d = 3$ corresponds to the three-component SU(2) KM on $S_2$. The solution for $\sigma_d(z)$ and $\Gamma_d(z)$ is given by:

$$\sigma_d(z) = \frac{z\Psi_d(z) - 1}{\Psi_d(z)}, \quad \Gamma_d(z) = \frac{z\Psi_d(z) - 1}{\Psi_d^2(z)}. \tag{135}$$

These equations can be obtained directly from the definition of $\sigma_d(z)$ and the Ward Identity

$$\sigma_d(z) = \Psi_d(z)\Gamma_d(z), \quad \Gamma_d(z) = \frac{1}{z^{d-1}}\frac{d}{dz}\left[z^{d-1}\Psi_d^{-1}(z)\right] = \mathrm{div}[\vec{v}_d], \quad \vec{v}_d = \Psi_d^{-1}\vec{e}_r. \tag{136}$$

Here we assumed that the GF has only "radial" component in d-dimensional spherical coordinate system.

The Dyson equations for the GF and SE are accompanied by the corresponding boundary conditions:

$$\frac{d\Psi_d(z)}{dz} + \left[z - \frac{d-1}{z}\right]\Psi_d(z) = 1, \quad \Psi_d(z \to \infty) = \frac{1}{z}, \tag{137}$$

$$\frac{d\sigma_d(z)}{dz} + \left[z + \frac{d-1}{z}\right]\sigma_d(z) - \sigma_d^2(z) = d, \quad \sigma_d(z \to \infty) = \frac{d}{z}. \tag{138}$$

The straightforward solution of the Dyson equation can be written by the Hilbert transform $H[u^{d-1}e^{-u^2}](z)$ (see Appendix B):

$$\Psi_d(z) = \frac{1}{2\Omega_d}\int_0^\infty u^{d-1}e^{-u^2/2}du\left[\frac{1}{z-u} + \frac{1}{z+u}\right]. \tag{139}$$

Here $\Omega_d = 2^{\frac{d}{2}-1}\Gamma(\frac{d}{2})$ is the normalizing coefficient and $\Gamma(u)$ is the Euler's Gamma-function. The $(d-1)/z$ "centrifugal" term is responsible for the level repulsion.

The differential identities connecting GF and SE are given by:

$$\sigma_d(z) = -\frac{d}{dz}\log\left[\frac{\Psi_d(z)}{z^{d-1}}\right], \quad \Gamma_d(z) = 1 - \frac{d\sigma_3(z)}{dz} = 1 + \frac{d^2}{dz^2}\log\left[\frac{\Psi_d(z)}{z^{d-1}}\right]. \tag{140}$$

These identities follow from the solution of Riccati-type equation for $\sigma_d(z)$.

**Large $D$-limit**

Differential equations describing the single-particle Green's function of the Keldysh model with a few-component Gaussian random potential contain coefficients $O(1)$. Keldysh models with a $D + 1 \gg 1$-component random potential contain a natural small parameter $1/D$ which can be used to find an approximate but reliable solution based on the $1/D$ expansion. To obtain the differential equation in the form convenient for the expansion we first proceed with re-scaling of the functions and the arguments: new GF $\phi = \Psi_D/\sqrt{D}$, new self-energy $\rho = \sigma_D/\sqrt{D}$ and new variable $w = z/\sqrt{D}$.

The differential equations for new GF and SE read:

$$\frac{1}{D}\frac{d\phi(w)}{dw} + \left[w - \frac{1}{w}\right]\phi(w) = \frac{1}{D}, \quad \phi(w \to \infty) = \frac{1}{Dw}, \tag{141}$$

$$\frac{1}{D}\frac{d\rho(z)}{dw} + \left[w + \frac{1}{w}\right]\rho(w) - \rho^2(w) = 1 + \frac{1}{D}, \quad \rho(w \to \infty) = \frac{1+1/D}{w}. \tag{142}$$

The Ward Identity acquires the form:

$$\Gamma(w) = \frac{1}{w^D}\frac{d}{dw}\left[w^D\phi^{-1}(w)\right] = \mathrm{div}\left[\vec{v}(w)\right], \quad \vec{v}(w) = \phi^{-1}(w)\vec{e}_r. \tag{143}$$

It is straightforward now to find an approximate solution of these equations. In particular, one can see that the boundary condition gives a reliable approximation for both the GF and SE at $z > D$:

$$\Psi_D^R(z) = \frac{1}{z - D/z + i\delta}, \qquad \sigma_D(z) = \frac{D}{z}. \tag{144}$$

The imaginary part of the $\Psi_D^R(z)$ in this approximation consists of two $\delta$-functions separated by $\Delta z = 2\sqrt{D}$.

To summarize, adding components of the random Gaussian field allows to engineer the gap in the DOS: few-component Keldysh models are characterized by a "soft" (pseudo) gap which becomes "harder" when the number of components increases.

# 7 *Étude* No 5. Perturbative expansion for the bold Green's function

The three last advanced *Études* are devoted to combinatorics of the Feynman diagrams for the bold Green's function and derivation of recurrence relations allowing to find a number of skeleton Feynman diagrams in a given order of the perturbation expansion. The "skeleton" means that we are looking for irreducible diagrams containing the bold GF (see e.g. Fig 12). As it has been mentioned before, the advantage of slow non-Markovian classical random potential models is that all diagrams in a given order of the perturbation theory have the same value. Thus, the coefficients of the expansion give the total number of the diagrams.[12] *Étude* 6 is based on the works of Sadovskii - Kuchinskii [4, 12] and Suslov [13] (SKS) who considered the enumeration of skeleton diagrams for the "orthodox" single-component Keldysh model. In this *Étude* we present a simple pedagogical derivation of the SKS recurrence equation using a method different from the original SKS works. This method proves to be very convinient for deriving a general equation giving access to enumeration of skeleton diagrams for classical random Gaussian field models with arbitrary number of the random field components. These equations will be derived and discussed in *Étude* 7. We will show that the SKS equations is just a particular limit of the general theory.

In this preparatory *Étude* we create a basis for answering questions about the combinatorics of Feynman diagrams. The first question is: "How many Feynman diagrams exist in n-th order of the perturbative expansion?". Or, put more simply "How many Feynman diagrams contain n-wavy lines for $d$-color random Gaussian field models?" This question is trivial as we already know the answers for one-, two- and three- component models. To answer this question we need to extend the general solution for the $d$-color GF $\Psi_d$ in terms of the bare GF in the limit $z \to \infty$:

$$\Psi_d(z) = \frac{1}{\Omega_d} \int_0^\infty \sum_{n=0}^\infty \frac{u^{2n+d-1}}{z^{2n+1}} e^{-u^2/2} du = \frac{1}{\Omega_d} \sum_{n=0}^\infty \frac{1}{z^{2n+1}} \int_0^\infty u^{2n+d-1} e^{-u^2/2} du = \sum_{n=0}^\infty \frac{A_n^{(d)}}{z^{2n+1}}.$$

Using equation for $\Omega_d = 2^{\frac{d}{2}-1}\Gamma(\frac{d}{2})$ from the previous *Étude* we get

$$A_n^{(d)} = \frac{2^n \Gamma\left(n + \frac{d}{2}\right)}{\Gamma\left(\frac{d}{2}\right)}. \tag{145}$$

This coefficient defines the number of Feynman diagrams in the $n$-th order of GF expansion.

---

[12]The situation is completely different for fast (Markovian) random potentials. In this case, the selection rule says that the rainbow-type diagrams give much large contribution in comparison to the diagrams with self-crossing.

$$\Sigma(\varepsilon) = \quad + \quad + \quad \cdots$$

Figure 12: Diagrammatic expansion for the self-energy. Double line corresponds to bold Green's function. Wavy line stands for the correlator of Gaussian classical random field.

To check that the general answer is correct, we write down explicitly the values of $A_n^{(d)}$ for $d = 1, 2, 3$:

$$A_n^{(1)} = (2n-1)!! , \quad A_n^{(2)} = (2n)!! , \quad A_n^{(3)} = (2n+1)!! . \tag{146}$$

Another important check is the number of diagrams in the lowest (first) order of the perturbative expansion:

$$A_1^{(d)} = \frac{2\Gamma\left(1 + \frac{d}{2}\right)}{\Gamma\left(\frac{d}{2}\right)} = d . \tag{147}$$

Finally, for the limit $d \gg 1$:

$$A_n^{(n \ll d/2)} \to 2^n . \tag{148}$$

The numbers given by Eq. (145) being plugged into the equations for the self-energies and the triangular vertex give the answer about the corresponding combinatorics in terms of bare GF.

## 8 *Étude* No 6. Enumeration of "skeleton" Feynman diagrams in a single-component Keldysh model: square recursion

Summation and re-summation methods of Feynman diagrams are widely used in analytical (quantum field theories) and numerical (bold diagrammatic Quantum Monte-Carlo methods, see, e.g. [19–21]) solutions in various problems of modern many-body physics. Counting the number of Feynman diagrams is another challenging problem, which is important, for example, for benchmarking numerical techniques based on the diagrammatic expansion. Hedin in his pioneering work [14] formulated a set of equations (Hedin's equations) connecting the single particle propagator $G$ with the effective potential $W$, self-energy $\Sigma$, polarization $\Pi$ and vertex $\Gamma$. Another additional equation follows from the functional-derivative connection between $\Gamma$, $\Sigma$ and $G$. In zero dimension of the space-time the Hedin equations become algebraic. By searching the solution of the Hedin equations in powers of the bare interacting potential and taking into account the number of closed loops, one gets the number of Feynman diagrams from the expansion coefficients [15–17]. Further development of these methods made it possible to formulate an iterative algorithm for counting Feynman diagrams via many-body relations [18] .

In this *Étude* we develop analytical methods for counting skeleton Feynman diagrams for self-energy, triangular vertex and single-particle T-matrix for models with $d$-component classical (no loops) Gaussian random field. We are dealing with $0 + 1$ models and therefore will end up with some differential (not algebraic, contrast to the Hedin's theory) equations. The methods discussed in *Études* 6 *and* 7 can also be generalized to enumerate diagrams for the two-particle Green's function and the full four-point correlator. In order to find the combinatorics for the self-energies and the triangular vertices of the single-component Keldysh

model in terms of the bold GF we start with the general equations Eq.(140) formally defining $\sigma_1$ and $\Gamma_1$ as Taylor series in terms of the bold GF $\Psi_1$:

$$\sigma_1[\Psi_1] \;=\; \frac{z[\Psi_1]\Psi_1-1}{\Psi_1} = \sum_{n=0}^{\infty} a_n^{(1)}\Psi_1^{2n+1}, \tag{149}$$

$$\Gamma_1[\Psi_1] \;=\; \frac{\sigma_1[\Psi_1]}{\Psi_1} = \frac{z[\Psi_1]\Psi_1-1}{\Psi_1^2} = \sum_{n=0}^{\infty} a_n^{(1)}\Psi_1^{2n}. \tag{150}$$

Thus, the goal is to find $z[\Psi_1]$. Since $\Psi_1(z)$ is expressed in terms of the Dawson function, we aim to construct the inverse Dawson function [11]. The coefficients $a_n^{(1)}$ in the asymptotic series fully determine the number of irreducible Feynman diagrams for the self-energy and the vertex correspondingly.

Using equation (149) we write:

$$z[\Psi_1] = \frac{1}{\Psi_1} + \sigma_1[\Psi_1] = \frac{1}{\Psi_1} + \sum_{n=0}^{\infty} a_n^{(1)}\Psi_1^{2n+1}. \tag{151}$$

We outline a simple and elegant way to construct the inverse Dawson function below. The function $z[\Psi_1] = \mathrm{invD}_+[\Psi_1]$ satisfies the equation

$$\frac{dz[\Psi_1]}{d\Psi_1}(1-\Psi_1 z[\Psi_1]) = 1, \tag{152}$$

which is obtained by inversion of Eq. (39). The boundary condition for this equation reads $\Psi_1 \cdot z[\Psi_1]|_{z\to\infty} = 1$. Substituting expansion for $z[\Psi_1]$ into (152) we get:

$$\left[1-\sum_{n=0}^{\infty}(2n+1)a_n^{(1)}\Psi_1^{2n+2}\right]\left[\sum_{n=0}^{\infty}a_n^{(1)}\Psi_1^{2n}\right] = 1, \tag{153}$$

which after opening the brackets is re-written as follows:

$$a_0^{(1)}+\sum_{n=1}^{\infty}a_n^{(1)}\Psi_1^{2n}-\sum_{n=1}^{\infty}\Psi_1^{2n}\sum_{m=0}^{n-1}[2(n-m)-1]a_m^{(1)}a_{n-m-1}^{(1)} = 1. \tag{154}$$

Equating the coefficients in front of the $n$-th power of the bold GF $\Psi_1^n$ we obtain the recurrence relations for the coefficients $a_n^{(1)}$:

$$a_n^{(1)} = \sum_{m=0}^{n-1}[2(n-m)-1]a_m^{(1)}a_{n-m-1}^{(1)}, \quad a_0^{(1)} = 1. \tag{155}$$

The equation (155) can be written in more compact form

$$\boxed{a_n^{(1)} = n\sum_{m=0}^{n-1}a_m^{(1)}a_{n-m-1}^{(1)}, \quad a_0^{(1)} = 1.} \tag{156}$$

The number of the irreducible diagrams in the first ten orders of the perturbation theory is presented in the Table 1.

The recurrence relations for the diagram combinatorics of the single-component Keldysh model have been obtained by Sadovskii and Kuchinskii [12] and Suslov [13] using a bit different technique. The large-n asymptotic of the coefficients $a_n^{(1)}$ has been analysed by Sadovskii

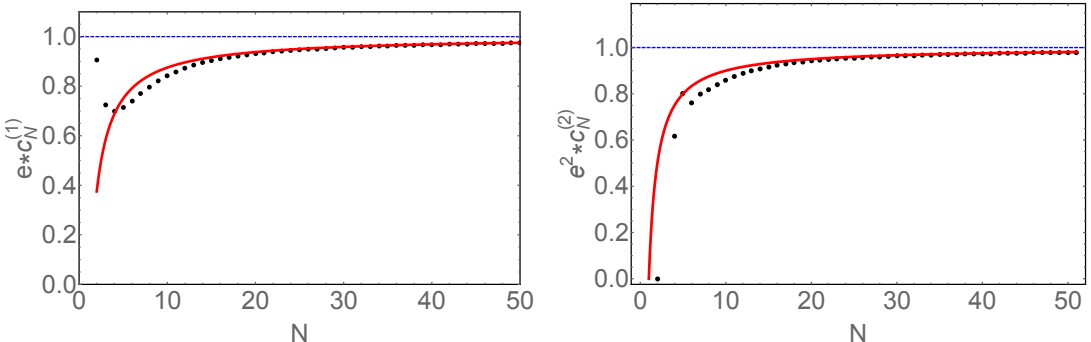

Figure 13: Asymptotic of the skeleton Feynman diagrams number for the self-energy for $d = 1$ KM model, red solid line is $1 - 5/(4n)$ (left) and $d = 2$ KM model, red solid line is $1 - 1/n$ (right).

and Kuchinskii in [12]. In the limit $n \gg 1$ SK transferred the recurrence equation (156) into an ordinary differential equation. Its solution gives the asymptotic:

$$a_n^{(1)} \xrightarrow{n \gg 1} \frac{1}{e} \left[ 1 - \frac{5}{4n} + O\left(\frac{1}{n^2}\right) \right] (2n + 1)!! \, . \tag{157}$$

We plot in Fig. 13 (left panel) the normalized number of irreducible Feynman diagrams for the first $n = 50$ orders of the perturbation theory (dots). The red solid line shows asymptotic behaviour given by Eq. (157) accounting for the $1/n$ correction.

The SK method, while powerful enough to handle $1/n$ corrections, does not, however, allow the coefficient $(1/e)$ before $(2n + 1)!!!$ to be determined analytically. This coefficient was calculated numerically by SK and derived analytically by Suslov [13]. In the next and the last *Étude* we will derive the recurrence relations for the self-energy and the triangular vertex combinatorics considering the multi-component Keldysh model. We will derive general recurrence equations which give the number of the skeleton diagrams in any order of expansion. To illustrate how to get the asymptotic behaviour of the coefficients of $d$-component models at the limit $n \gg 1$ we apply Sadovskii and Kuchinskii method.

# 9 *Étude* No 7. Enumeration of "skeleton" Feynman diagrams in a multi-component Keldysh model: cubic recursion

This *Étude* begins with derivation of the recurrence equation for the two-component Keldysh model. Following the key steps of the derivation we present detailed discussion for computing the large - $n$ asymptotic for the number of skeleton diagrams. After "learning" how the method works for a two-component random Gaussian field we proceed with the general derivation for the multi-component case. At the end of this *Étude* we briefly discuss how the combinatorics of skeleton diagrams for self-energy can be used for the combinatorics of the diagrams for the $T$-matrix.

**Enumeration of "skeleton" diagrams for the two-component Keldysh model**

The differential equation for the inverted function $z[\Psi_2]$ is obtained by inversion the differential equation for $\Psi_2(z)$:

$$\frac{dz[\Psi_2]}{d\Psi_2} \left( z[\Psi_2] - \Psi_2 \left[ (z[\Psi_2])^2 - 1 \right] \right) = z[\Psi_2] \, .$$

We substitute the Taylor expansion for $\sigma_2[\Psi_2]$

$$z[\Psi_2] = \frac{1}{\Psi_2} + \sigma_2[\Psi_2] = \frac{1}{\Psi_2} + \sum_{n=0}^{\infty} a_n^{(2)} \Psi_2^{2n+1}, \qquad (158)$$

to get the recurrence relations for the coefficients $a_n^{(2)}$:

$$\left[ 1 - \sum_{n=0}^{\infty} (2n+1) a_n^{(2)} \Psi_2^{2n+2} \right] \left[ \sum_{n=0}^{\infty} a_n^{(2)} \Psi_2^{2n} - 1 + \Psi_2^2 \left( \sum_{n=0}^{\infty} a_n^{(2)} \Psi_2^{2n} \right)^2 \right] = 1 + \sum_{n=0}^{\infty} a_n^{(2)} \Psi_2^{2n+2}.$$

This equation finally leads to the following recurrence relations:

$$a_0^{(2)} = 2, \qquad a_1^{(2)} = 0, \qquad (159)$$

$$a_n^{(2)} = -2(n-1) a_{n-1}^{(2)} + 2 \sum_{m=0}^{n-1} [n-m-1] a_m^{(2)} a_{n-m-1}^{(2)}$$

$$+ \sum_{m=0}^{n-2} \sum_{p=0}^{n-m-2} [2(n-m-p-2)+1] a_m^{(2)} a_p^{(2)} a_{n-m-p-2}^{(2)}.$$

We notice an interesting property of the case $d = 2$, namely, that the coefficient $a_1^{(2)} = 0$. It reflects the fact that the second order irreducible Feynman diagram vanishes due to topology of the two-component model (see *Étude* 2). The equation (160) can be further simplified:

$$\boxed{a_n^{(2)} = -2(n-1) a_{n-1}^{(2)} + (n-1) \sum_{m=0}^{n-1} a_m^{(2)} a_{n-m-1}^{(2)} + \sum_{m=0}^{n-2} \sum_{p=0}^{n-m-2} [n-m-1] a_m^{(2)} a_p^{(2)} a_{n-m-p-2}^{(2)}.}$$

This recurrence equation generalizes the combinatorics of the Keldysh model to the case of the two-component random Gaussian field. The main difference from SKS equation is that in addition to the bilinear recurrence, a cubic term appears in the recurrence equation.

Table 1: Expansion coefficients for the self-energy dependence on the bold GF for $d = 1, 2, 3$ KM.

| $N$ | $a_{N-1}^{(1)}$ | $a_{N-1}^{(2)}$ | $b_N^{(2)} = a_{N-1}^{(2)}/2^N$ | $a_{N-1}^{(3)}$ |
|---|---|---|---|---|
| 1 | 1 | 2 | 1 | 3 |
| 2 | 1 | 0 | 0 | -3 |
| 3 | 4 | 8 | 1 | 24 |
| 4 | 27 | 32 | 2 | -45 |
| 5 | 248 | 416 | 13 | 1044 |
| 6 | 2830 | 4736 | 74 | 3078 |
| 7 | 38232 | 69632 | 544 | 119232 |
| 8 | 593859 | 1141248 | 4458 | 1517427 |
| 9 | 10401712 | 21105152 | 41221 | 33013980 |
| 10 | 202601898 | 431525888 | 421412 | 670457250 |
| $N \gg 1$ | $\frac{1}{e}(2N-1)!!$ | $\frac{1}{e^2}(2N)!!$ | $\frac{1}{e^2}N!$ | $\frac{1}{e^3}(2N+1)!!$ |

To find the asymptotic behaviour of $a_n^{(2)}$ at $n \gg 1$ we will follow the method developed by Sadovskii and Kuchinskii in [12]. We look for the asymptotic solution of (160) in the form

$$a_n^{(2)} \approx (\alpha n + \beta) a_{n-1}^{(2)}. \tag{160}$$

Substituting Eq. (160) into Eq. (160) we find that at the limit $n \gg 1$

$$\alpha = \beta = 2. \tag{161}$$

We can therefore re-parametrize Eq. (160) by introducing the new coefficients as follows:

$$a_n^{(2)} = 2^{n+1}(n+1)! c_n^{(2)}. \tag{162}$$

This re-parametrization gives the asymptotic behaviour for the number of the skeleton diagrams for the self-energy in the large-$n$ limit. In order to find the $1/n$ correction we need to deal with the equations for the coefficients $c_n^{(2)}$. By plugging in Eq. (162) into Eq. (160) we get a new recurrence relations:

$$c_n^{(2)} = -\frac{n-1}{n+1} c_{n-1}^{(2)} + \frac{n-1}{2} \sum_{m=0}^{n-1} \frac{(m+1)!(n-m-1)!}{(n+1)!} c_m^{(2)} c_{n-m-1}^{(2)}$$

$$+ \sum_{m=0}^{n-2} \sum_{p=0}^{n-m-2} [n-m-1] \frac{(m+1)!(p+1)!(n-p-m-2)!}{(n+1)!} c_m^{(2)} c_p^{(2)} c_{n-m-p-2}^{(2)}.$$

Taking the limit of large $n$ and retaining only the leading $1/n^2$ terms in (163) under assumption $c = c_n \approx c_{n-1} \approx c_{n-2}$ we finally obtain a simple differential equation:

$$\frac{dc}{dn} = \frac{1}{n^2} c. \tag{163}$$

The solution of this equation is given by

$$c = \text{const} \cdot \exp\left(-\frac{1}{n}\right) \xrightarrow{n \gg 1} \text{const}\left(1 - \frac{1}{n} + O\left(\frac{1}{n^2}\right)\right). \tag{164}$$

As it has been discussed before, Sadovskii-Kuchinskii method does not provide information about the coefficient in front of the exponential function. To obtain correctly this coefficient we follow the logic of Suslov's work [13]. Let's begin with $d = 1$ case. For all $d \neq 1$, since the noise components are independent (no cross-correlations), we can repeat the same arguments for each component independently and hence the constant found for the $d = 1$ case will be replaced by the constant to the power $d$. First, we observe that the coefficient $B_n^{(1)} = a_{n-1}^{(1)} = (2n)!! c_{n-1}^{(1)}$ has the same $n$-dependence as the coefficient $A_n^{(1)}$ which defines the total number of the Feynman diagrams for GF[13]. Second, we see that the diagrams contributing to $A_n^{(1)}$ consist of all irreducible diagrams contributing to $B_n^{(1)}$ plus all reducible diagrams $B_{n-1}^{(1)} = B_n^{(2)}/1!$ with one wavy line insert. Third, there are diagrams of order $n-2$ with $B_{n-2}^{(1)} = B_n^{(2)}/2!$ and so on. As a result,[14] we get the relation:

$$A_n^{(1)} = B_n^{(1)}\left(1 + \frac{1}{1!} + \frac{1}{2!} + ...\right) \xrightarrow{n \gg 1} e B_n^{(1)}. \tag{165}$$

---

[13]It is easy to see that the number of skeleton diagrams in the $n$-th order of the perturbation theory is defined strictly speaking by the coefficients $B_n^{(1)}$ while the total number of diagrams is defined by $A_n^{(1)}$.

[14]We omit $O(1/n!)$ corrections to the constant in the large $n$ limit.

Similarly, for $d = 2$ we obtain

$$A_n^{(2)} = B_n^{(2)}\left(1 + \frac{1}{1!} + \frac{1}{2!} + ...\right)^2 \xrightarrow{n \gg 1} e^2 B_n^{(2)}.$$ (166)

Finally, using const $= 1/e^2$ we get:

$$a_n^{(2)} \xrightarrow{n \gg 1} \frac{1}{e^2}\left[1 - \frac{1}{n} + O\left(\frac{1}{n^2}\right)\right]2^{n+1}(n+1)!.$$ (167)

Note that for $d = 2$ each random Gaussian field line gives the factor 2 because the correlator $\langle \tau^+ \tau^- \rangle = \langle \tau^x \tau^x \rangle + \langle \tau^y \tau^y \rangle$ and the correlations in the $x$- and $y$- pseudospin directions are statistically independent. The second column of the Table 1 contains coefficient $a_{N-1}^2$ computed for the first ten orders of the perturbation theory. The number of irreducible diagrams in the $n$-th order is given by the coefficient $b_N^{(2)} = a_{N-1}^{(2)}/2^N$. Note that the coefficient $b_2^{(2)} = 0$ since the second order irreducible Feynman diagram is absent (see $Étude$ 2).

We plot in Fig. 13 (right panel) the normalized number of irreducible Feynman diagrams for the first $n = 50$ orders of the perturbation theory (dots). The red solid line shows the asymptotic behaviour given by Eq. (167) accounting for the $1/n$ correction.

Before proceeding to the discussion of the general $d$-component combinatorics, let us make an interesting remark. In the last column of the Table 1 we present the coefficients $a_{N-1}^3$ for the $d = 3$ component Keldysh model. For the two previously considered cases, namely, $d = 1$ and $d = 2$ we observe that all coefficients $a_{N-1}^{d<3}$ are positive. For $d = 3$, however, we see that the coefficients for $N = 2$ and $N = 4$ become negative. This is the generic situation for Keldysh models with $d > 2$. Since the coefficients $a_{N-1}^{d>2}$ determine not only the number of the skeleton diagrams, but also the total sign, we briefly illustrate the source of the negative sign in Fig. 14.

**General $d > 1$ - component isotropic Keldysh model**

Let us now consider a general multi-component Keldysh model with the number of Gaussian random field components $d > 1$. The differential equation for the inverse function $z[\Psi_d]$ is given by

$$\frac{dz[\Psi_d]}{d\Psi_d}\left(z[\Psi_d] - \Psi_d\left[(z[\Psi_d])^2 - (d-1)\right]\right) = z[\Psi_d].$$

Substituting the Taylor expansion for the self-energy in terms of the bold GF $\Psi_d$]

$$z[\Psi_d] = \frac{1}{\Psi_d} + \sigma_d[\Psi_d] = \frac{1}{\Psi_d} + \sum_{n=0}^{\infty} a_n^{(d)}\Psi_d^{2n+1},$$ (168)

we obtain recurrence relations between the coefficients $a_n^{(d)}$.

$$\left[1 - \sum_{n=0}^{\infty}(2n+1)a_n^{(d)}\Psi_d^{2n+2}\right]\left[\sum_{n=0}^{\infty} a_n^{(d)}\Psi_d^{2n} - (d-1) + \Psi_d^2\left(\sum_{n=0}^{\infty} a_n^{(d)}\Psi_d^{2n}\right)^2\right]$$ (169)

$$= 1 + \sum_{n=0}^{\infty} a_n^{(d)}\Psi_d^{2n+2}.$$

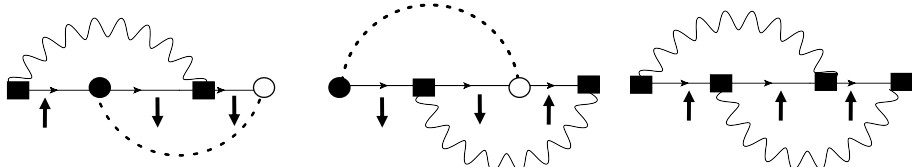

Figure 14: We illustrate why for $d > 2$ the coefficient $a_1^{(d)} = d(2-d) < 0$. Let us take an example $d = 3$. The Feynman diagrams contain wavy line denoting $z-z$ noise correlator and dashed line denoting $+- = xx + yy$ noise correlator. It is straightforward to show that three 3rd order irreducible diagrams come with factors $-2$ another $-2$ and $+1$ which gives finally overall factor $-3$. Sign $-$ is associated with the multiplication rule $\det[\tau^z] = \tau_{11}^z \tau_{22}^z = -1$.

Finally, following the steps outlined in the previous $\acute{E}tude$ we get:

$$a_0^{(d)} = d, \qquad a_1^{(d)} = d(2-d), \tag{170}$$

$$a_n^{(d)} = -\{(d-1)(2n-1)-1\}a_{n-1}^{(d)} + 2\sum_{m=0}^{n-1}[n-m-1]a_m^{(d)}a_{n-m-1}^{(d)}$$

$$+ \sum_{m=0}^{n-2}\sum_{p=0}^{n-m-2}[2(n-m-p-2)+1]a_m^{(d)}a_p^{(d)}a_{n-m-p-2}^{(d)}.$$

The last equation can be re-written as follows:

$$\boxed{\begin{aligned} a_n^{(d)} = &-\{2n(d-1)-d\}a_{n-1}^{(d)} + (n-1)\sum_{m=0}^{n-1}a_m^{(d)}a_{n-m-1}^{(d)} \\ &+ \sum_{m=0}^{n-2}\sum_{p=0}^{n-m-2}[n-m-1]a_m^{(d)}a_p^{(d)}a_{n-m-p-2}^{(d)}. \end{aligned}} \tag{171}$$

This equation is the central point of this $\acute{E}tude$ and the central result of the paper.

The large-$n$ asymptotes and the $1/n$ expansion for the coefficients $a_n^{(d)}$ can be straightforwardly performed using the Sadovskii-Kuchinskii-Suslov [4,12,13] method which leads to the following final result:

$$\boxed{a_n^{(d)} \xrightarrow{n \gg 1} \frac{1}{e^d}\left[1 - \frac{f(d)}{n} + O\left(\frac{1}{n^2}\right)\right]\frac{2^{n+1}\Gamma(n+1+d/2)}{\Gamma(d/2)},} \tag{172}$$

where the function $f(d)$ is computed in the previous subsection for two particular cases: $f(d=1) = 5/4$ and $f(d=2) = 1$. We leave the problem of determining $f(d)$ for arbitrary $d$ as an exercise for readers.

## Enumeration of "skeleton" diagrams for the T-matrix

The T-matrix is related to the self-energy by the following relation: $T_d \cdot G_0 = \Sigma \cdot G$. Equivalently, for the dimensionless T-matrix function of the dimensionless variable $z$, $\mathbb{T}_d \cdot z^{-1} = \sigma_d \cdot \Psi_d$:

$$\mathbb{T}_d(z) = z\Psi_d(z)\sigma_d(z) = z^2\Psi_d(z) - z. \tag{173}$$

Combinatorics of the diagrams for the T-matrix is defined through the equation:

$$\boxed{\mathbb{T}_d[\Psi_d] = \sum_{n=0}^{\infty}a_n^{(d)}\Psi_d^{2n+1} + \sum_{n=0}^{\infty}\sum_{m=0}^{\infty}a_n^{(d)}a_m^{(d)}\Psi_d^{2(n+m)+3}.} \tag{174}$$

Therefore, there is no need to construct independent equations for the T-matrix combinatorics since it can be determined from the number given by the coefficients $a_n^{(d)}$:

$$\mathcal{T}_d[\Psi_d] = \sum_{n=0}^{\infty}\left[a_n^{(2)} + \sum_{m=0}^{\infty} a_m^{(d)}\Psi_d^{2(m+1)}\right]\Psi_d^{2n+1} = \sum_{n=0}^{\infty} r_n^{(d)}\Psi_d^{2n+1}, \tag{175}$$

with

$$r_n^{(d)} = a_n^{(d)} + \sum_{m=0}^{n-1} a_m^{(d)} a_{n-m-1}^{(d)}. \tag{176}$$

Thus, the combinatorics of the coefficients $a_n^{(d)}$ for the self-energy fully determines the combinatorics of the vertex part and the single particle T-matrix.

## 10 *Coda*. Concluding remarks

To summarize, let us discuss what we can learn from playing *Seven Études* on dynamical Keldysh model.

- The single particle Green's function of the Keldysh models with a $d$-component non-Markovian classical random Gaussian field satisfies the first order ordinary linear differential equation. The Dyson equations can be obtained in the closed form due to the Ward Identities and solved exactly for an arbitrary number of the random field components.

- The exact solution for the single-particle Green's function corresponds to an ensemble Gaussian averaging of the single particle Green's function over the Gaussian realizations of the random potential.

- The imaginary part of the single particle GF (proportional to the Density of States) has the Gaussian shape.

- If the number of components is $d > 1$, there exists a "centrifugal" term in the Dyson equation responsible for level repulsion. The level repulsion becomes more pronounced when the number component of the random fields grows.

- All Keldysh models with even numbers of components contain intrinsic non-analyticity in the single particle correlators in the zero frequency limit due to the Rayleigh-type distribution. The single particle correlators of the Keldysh models with odd numbers of random field components are analytic at all frequencies due to the Gaussian distribution.

- The total number of skeleton Feynman diagrams in a given order of expansion with respect to a multi-component random potential satisfy some recurrence equations containing linear, bilinear and cubic terms. The single component Keldysh model has some additional degeneracy as both linear and cubic terms are absent.

- The combinatorics of the skeleton Feynman diagrams for the self-energy completely determines the corresponding combinatorics of the full vertices and the single-particle T-matrices.

Multi-component Keldysh models describe the behaviour of complex (double, triple- etc) quantum dots subject to slowly fluctuating electric potentials which determine the shape of the quantum wells and inter-dot barriers. The application of the Keldysh model to computing dynamical correlation functions of many-body Fermi and Bose systems, as well as some other realizations of the Keldysh model in theories with multi-component synthetic gauge fields [22], opens interesting directions for future research.

## Acknowledgements

The Authors are thankful to late Kostya Kikoin for many years of fruitful collaboration on numerous topics of modern condensed matter physics including but not limited by dynamical symmetries and the Keldysh model. We are grateful to Boris Altshuler for very fruitful discussions on disordered systems and Nikolay Prokof'ev and Boris Svistunov for teaching the basics of the bold diagrammatic Quantum Monte Carlo method. DE thanks DFG (project number 405940956) for the partial financial support. The work of MK is conducted within the framework of the Trieste Institute for Theoretical Quantum Technologies (TQT).

## A   Mathematical Instrumentation

### Gell-Mann matrices

The Gell-Mann matrices constitute the fundamental representation of the group $SU(3)$. The eight generators satisfy the commutation relations:

$$[\Lambda_\alpha, \Lambda_\beta] = i f^\lambda_{\alpha\beta\gamma} \Lambda_\gamma, \tag{A1}$$

$$\{\Lambda_\alpha, \Lambda_\beta\} = \frac{1}{3}\delta_{\alpha\beta}\mathbb{I} + d^\lambda_{\alpha\beta\gamma}\Lambda_\gamma. \tag{A2}$$

We refer to the conventional form of the Gell-Mann matrices as the $\lambda$-basis, $\Lambda_\alpha = \lambda_\alpha/2$:

$$
\lambda_1 = \begin{pmatrix} 0 & 1 & 0 \\ 1 & 0 & 0 \\ 0 & 0 & 0 \end{pmatrix}, \qquad
\lambda_2 = \begin{pmatrix} 0 & -i & 0 \\ i & 0 & 0 \\ 0 & 0 & 0 \end{pmatrix}, \quad
\lambda_3 = \begin{pmatrix} 1 & 0 & 0 \\ 0 & -1 & 0 \\ 0 & 0 & 0 \end{pmatrix},
$$
$$
\lambda_4 = \begin{pmatrix} 0 & 0 & 1 \\ 0 & 0 & 0 \\ 1 & 0 & 0 \end{pmatrix}, \qquad
\lambda_5 = \begin{pmatrix} 0 & 0 & -i \\ 0 & 0 & 0 \\ i & 0 & 0 \end{pmatrix}, \quad
\lambda_6 = \begin{pmatrix} 0 & 0 & 0 \\ 0 & 0 & 1 \\ 0 & 1 & 0 \end{pmatrix}, \tag{A3}
$$
$$
\lambda_7 = \begin{pmatrix} 0 & 0 & 0 \\ 0 & 0 & -i \\ 0 & i & 0 \end{pmatrix}, \quad
\lambda_8 = \frac{1}{\sqrt{3}}\begin{pmatrix} 1 & 0 & 0 \\ 0 & 1 & 0 \\ 0 & 0 & -2 \end{pmatrix}.
$$

The $\lambda$- matrices are normalized by the identity

$$Tr(\lambda_\alpha \lambda_\beta) = 2\delta_{\alpha\beta}. \tag{A4}$$

The structure constants are

$$f^\lambda_{\alpha\beta\gamma} = \frac{1}{4i}\mathrm{Tr}([\lambda_\alpha, \lambda_\beta] \cdot \lambda_\gamma), \tag{A5}$$

$$d^\lambda_{\alpha\beta\gamma} = \frac{1}{4}\mathrm{Tr}(\{\lambda_\alpha, \lambda_\beta\} \cdot \lambda_\gamma). \tag{A6}$$

It is convenient to use the rotated basis of the Gell-Mann matrices [9]. The rotation is performed to embed the representation $S = 1$ of the group $SU(2)$ as first three generators of the group $SU(3)$. We refer to the rotated representation of Gell-Mann matrices as the $\mu$- basis, $M_\alpha = \mu_\alpha/2$:

$$[M_\alpha, M_\beta] = i f^\mu_{\alpha\beta\gamma} M_\gamma, \tag{A7}$$

$$\{M_\alpha, M_\beta\} = \frac{1}{3}\delta_{\alpha\beta}\mathbb{I} + d^\mu_{\alpha\beta\gamma}M_\gamma, \tag{A8}$$

where

$$\mu_1 = \frac{1}{\sqrt{2}}\begin{pmatrix} 0 & 1 & 0 \\ 1 & 0 & 1 \\ 0 & 1 & 0 \end{pmatrix}, \qquad \mu_2 = \frac{1}{\sqrt{2}}\begin{pmatrix} 0 & -i & 0 \\ i & 0 & -i \\ 0 & i & 0 \end{pmatrix}, \qquad \mu_3 = \begin{pmatrix} 1 & 0 & 0 \\ 0 & 0 & 0 \\ 0 & 0 & -1 \end{pmatrix},$$

$$\mu_4 = \begin{pmatrix} 0 & 0 & 1 \\ 0 & 0 & 0 \\ 1 & 0 & 0 \end{pmatrix}, \qquad \mu_5 = \begin{pmatrix} 0 & 0 & -i \\ 0 & 0 & 0 \\ i & 0 & 0 \end{pmatrix}, \qquad \mu_6 = -\frac{1}{\sqrt{2}}\begin{pmatrix} 0 & 1 & 0 \\ 1 & 0 & -1 \\ 0 & -1 & 0 \end{pmatrix},$$

$$\mu_7 = -\frac{1}{\sqrt{2}}\begin{pmatrix} 0 & -i & 0 \\ i & 0 & i \\ 0 & -i & 0 \end{pmatrix}, \qquad \mu_8 = -\frac{1}{\sqrt{3}}\begin{pmatrix} 1 & 0 & 0 \\ 0 & -2 & 0 \\ 0 & 0 & 1 \end{pmatrix}.$$

The $\mu$- matrices are normalized by identity

$$Tr(\mu_\alpha\mu_\beta) = 2\delta_{\alpha\beta}. \tag{A9}$$

The structure constants are

$$f^\mu_{\alpha\beta\gamma} = \frac{1}{4i}\mathrm{Tr}([\mu_\alpha, \mu_\beta]\cdot\mu_\gamma), \tag{A10}$$

$$d^\mu_{\alpha\beta\gamma} = \frac{1}{4}\mathrm{Tr}(\{\mu_\alpha, \mu_\beta\}\cdot\mu_\gamma). \tag{A11}$$

The connections between two basis are

$$\mu_1 = (\lambda_1 + \lambda_6)/\sqrt{2}, \quad \mu_2 = (\lambda_2 + \lambda_7)/\sqrt{2}, \quad \mu_3 = (\sqrt{3}\lambda_8 + \lambda_3)/2, \quad \mu_4 = \lambda_4,$$
$$\mu_5 = \lambda_5, \quad \mu_6 = -(\lambda_1 - \lambda_6)/\sqrt{2}, \quad \mu_7 = -(\lambda_2 - \lambda_7)/\sqrt{2}, \quad \mu_8 = (\lambda_8 - \sqrt{3}\lambda_3)/2. \tag{A12}$$

**Hilbert transform, Dawson function and exponential integral**

In this Instrumentation we summarize the equations for the Hilbert transform representation of the $d$- component Keldysh model Green's functions

- Odd $d = 2n + 1$ number of the random Gaussian field components
  The Hilbert transform $H_n[u^{2n}e^{-u^2}](z)$ is defined on a line $-\infty \infty$

$$H_n(z) = \frac{1}{\pi}\mathrm{P.V.}\int_{-\infty}^{\infty}\frac{u^{2n}e^{-u^2}}{z-u}du. \tag{B1}$$

Introducing

$$H^a(z) = \frac{1}{\pi}\mathrm{P.V.}\int_{-\infty}^{\infty}\frac{e^{-ax^2}}{z-x}dx = \frac{2}{\sqrt{\pi}}F(z\sqrt{a}), \tag{B2}$$

we relate $H_n[u^{2n}e^{-u^2}](z)$ to the Dawson function

$$H_n(z) = (-1)^n\left.\frac{\partial^n H^a(z)}{\partial a^n}\right|_{a=1}. \tag{B3}$$

Alternatively $H_n$ can be calculated using the recurrence relations

$$H_{n+1}(z) = z^2 H_n(z) - z\frac{(2n-1)!!}{2^n\sqrt{\pi}}. \tag{B4}$$

- Even $d = 2n$ number of the random Gaussian field components

$$\psi_n(z) = \frac{1}{2}\text{P.V.}\int_0^\infty u^{2n+1}e^{-u^2}du\left[\frac{1}{z-u} + \frac{1}{z+u}\right].\tag{B5}$$

We define

$$\psi^a(z) = \frac{1}{2}\text{P.V.}\int_0^\infty ue^{-au^2}du\left[\frac{1}{z-u} + \frac{1}{z+u}\right] = \frac{1}{2}z\,e^{-az^2}\text{Ei}\left[az^2\right].\tag{B6}$$

As a result we relate $\psi_n$ and Exponential Integral function derivatives

$$\psi_n(z) = (-1)^n\left.\frac{\partial^n\psi^a(z)}{\partial a^n}\right|_{a=1}.\tag{B7}$$

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
