# Peer review of "Seven Etudes on dynamical Keldysh Model"

_SciPost Physics Lecture Notes, doi:SciPost Phys. Lect. Notes 65 (2022)_

## Round 2 · Referee Report · Michael Sadovski (Referee 1) · 2022-9-14

Strengths

  1. Pedagogical style
  2. Clear presentation
  3. Important contribution to this field

Weaknesses

None that should be mentioned

Report

This paper presents an interesting and elegant discussion of a simplified dynamic field - theory models (dynamical generalization of the so called Keldysh model in the theory of disordered systems), which belongs to rather limited class of models where ALL Feynman diagrams (of perturbation series) can be summed exactly. Actually, the authors deal with a particle in a multicomponent dynamic, non-Markovian Gaussian random field. They present exact results for single-particle Green's functions, self-energies and vertex parts. Their results are based mainly on a closed form solution of the Dyson equation combined with the Ward identity. Asymptotically exact equations for the number of skeleton diagrams in the limit of large N are derived. Some examples of exact perturbation series summation are also presented. In this respect they analyze the combinatorics of the Feynman diagrams for the Green's function and the skeleton diagrams for the self-energy and vertex, using (and generalizing) the recurrence relations between Taylor expansion coefficients of self-energy, derived in earlier works by Kuchinskii, Sadovskii and Suslov. Possible physical realizations of a their multicomponent Gaussian random model in quantum transport via complex quantum dot experiments are also discussed in detail.

Requested changes

None

---

## Editorial Decision

published